# High-Dimensional Calibration from Swap Regret

**Maxwell Fishelson**[*]
maxfish@mit.edu

**Noah Golowich**[†]
nzg@mit.edu

**Mehryar Mohri**[‡]
mohri@google.com

**Jon Schneider**[§]
jschnei@google.com

## Abstract

We study the online calibration of multi-dimensional forecasts over an arbitrary convex set $\mathcal{P} \subset \mathbb{R}^d$ relative to an arbitrary norm $\|\cdot\|$. We connect this with the problem of external regret minimization for online linear optimization, showing that if it is possible to guarantee $O(\sqrt{\rho T})$ worst-case regret after $T$ rounds when actions are drawn from $\mathcal{P}$ and losses are drawn from the dual $\|\cdot\|_*$ unit norm ball, then it is also possible to obtain $\epsilon$-calibrated forecasts after $T = \exp(O(\rho/\epsilon^2))$ rounds. When $\mathcal{P}$ is the $d$-dimensional simplex and $\|\cdot\|$ is the $\ell_1$-norm, the existence of $O(\sqrt{T \log d})$-regret algorithms for learning with experts implies that it is possible to obtain $\epsilon$-calibrated forecasts after $T = \exp(O(\log d/\epsilon^2)) = d^{O(1/\epsilon^2)}$ rounds, recovering a recent result of [Pen25].

Interestingly, our algorithm obtains this guarantee without requiring access to any online linear optimization subroutine or knowledge of the optimal rate $\rho$ – in fact, our algorithm is identical for every setting of $\mathcal{P}$ and $\|\cdot\|$. Instead, we show that the optimal regularizer for the above OLO problem can be used to upper bound the above calibration error by a swap regret, which we then minimize by running the recent TreeSwap algorithm ([DDFG24, PR24]) with Follow-The-Leader as a subroutine. The resulting algorithm is highly efficient and plays a distribution over simple averages of past observations in each round.

Finally, we prove that any online calibration algorithm that guarantees $\epsilon T$ $\ell_1$-calibration error over the $d$-dimensional simplex requires $T \geq \exp(\mathrm{poly}(1/\epsilon))$ (assuming $d \geq \mathrm{poly}(1/\epsilon)$). This strengthens the corresponding $d^{\Omega(\log 1/\epsilon)}$ lower bound of [Pen25], and shows that an exponential dependence on $1/\epsilon$ is necessary.

## 1 Introduction

Consider the problem faced by a forecaster who must report probabilistic predictions for a sequence of events (e.g. whether it will rain or not tomorrow). One of the most common methods to evaluate the quality of such a forecaster is to verify whether they are *calibrated*: for example, does it indeed rain with probability 40% on days where the forecaster makes this prediction? In addition to calibration being a natural property to expect from predictions, several applications across machine learning, fairness, and game theory require the ability to produce online calibrated predictions [ZME20, GPSW17, HJKRR18, FV97].

When events have binary outcomes, calibration can be quantified by the notion of *expected calibration error*, which measures the expected distance between a prediction made by a forecaster and the actual empirical probability of the outcome on the days where they made that prediction. In a seminal result by Foster and Vohra [FV98], it was proved that it is possible for an online forecaster to efficiently

---

[*]MIT.

[†]MIT. Supported by a NSF Graduate Research Fellowship and a Fannie & Hertz Foundation Graduate Fellowship.

[‡]Google Research and Courant Institute of Mathematical Sciences, New York.

[§]Google Research.

39th Conference on Neural Information Processing Systems (NeurIPS 2025).

guarantee a sublinear calibration error of $O(T^{2/3})$ against any adversarial sequence of $T$ binary events. Equivalently, this can be interpreted as requiring at most $O(\epsilon^{-3})$ rounds of forecasting to guarantee an $\epsilon$ per-round calibration error on average.

However, many applications require forecasting sequences of *multi-dimensional* outcomes. The previous definition of calibration error easily extends to the multi-dimensional setting where predictions and outcomes belong to a $d$-dimensional convex set $\mathcal{P} \subset \mathbb{R}^d$. Specifically, if a forecaster makes a sequence of predictions $p_1, p_2, \ldots, p_T \in \mathcal{P}$ for the outcomes $y_1, y_2, \ldots, y_T \in \mathcal{P}$, their $\|\cdot\|$-calibration error (for any norm $\|\cdot\|$ over $\mathbb{R}^d$) is given by

$$\mathsf{Cal}_T^{\|\cdot\|} = \sum_{t=1}^{T} \|p_t - \nu_{p_t}\|$$

where $\nu_{p_t}$ is the average of the outcomes $y_t$ on rounds where the learner predicted $p_t$.

The algorithm of Foster and Vohra extends to the multidimensional calibration setting, but at the cost of producing bounds that decay exponentially in the dimension $d$. In particular, their algorithm only guarantees that the forecaster achieves an average calibration error of $\epsilon$ after $(1/\epsilon)^{\Omega(d)}$ rounds. Until recently, no known algorithm achieved a sub-exponential dependence on $d$ in any non-trivial instance of multi-dimensional calibration.

In 2025, [Pen25] presented a new algorithm for high-dimensional calibration, demonstrating that it is possible to obtain $\ell_1$-calibration rates of $\epsilon T$ in $d^{O(1/\epsilon^2)}$ rounds for predictions over the $d$-dimensional simplex (i.e., multi-class calibration). In particular, this is the first known algorithm achieving polynomial calibration rates in $d$ for fixed constant $\epsilon$. [Pen25] complements this with a lower bound, showing that in the worst case $d^{\Omega(\log 1/\epsilon)}$ rounds are necessary to obtain this rate (implying that a fully polynomial bound $\mathrm{poly}(d, 1/\epsilon)$ is impossible).

## 1.1 Our results

Although the algorithm of [Pen25] is simple to describe, its analysis is fairly nuanced and tailored to $\ell_1$-calibration over the simplex (e.g., by analyzing the KL divergence between predictions and distributions of historical outcomes). We present a very similar algorithm (`TreeCal`) for multi-dimensional calibration over an arbitrary convex set $\mathcal{P} \subset \mathbb{R}^d$, but with a simple, unified analysis that provides simultaneous guarantees for calibration with respect to any norm $\|\cdot\|$. In particular, we prove the following theorem.

**Theorem 1.1** (Informal restatement of Corollary C.5). *Fix a convex set $\mathcal{P}$ and a norm $\|\cdot\|$. Assume there exists a function $R : \mathcal{P} \to \mathbb{R}$ that is 1-strongly-convex with respect to $\|\cdot\|$ and has range $(\max_{x \in \mathcal{P}} R(x) - \min_{p \in \mathcal{P}} R(x))$ at most $\rho$. Then* `TreeCal` *guarantees that the calibration error of its predictions is bounded by* $\mathsf{Cal}_T^{\|\cdot\|} \leq \epsilon T$ *for* $T \geq (\mathsf{diam}_{\|\cdot\|}(\mathcal{P})/\epsilon)^{O(\rho/\epsilon^2)}$.

Interestingly, the function $R(p)$ and parameter $\rho$ appearing in the statement of Theorem 1.1 have an independent learning-theoretic interpretation: if we consider the *online linear optimization* problem where a learner plays actions in $\mathcal{P}$ and the adversary plays linear losses that are unit bounded in the dual norm $\|\cdot\|_*$, then it is possible for the learner to guarantee a regret bound of at most $O(\sqrt{\rho T})$ by playing Follow-The-Regularized-Leader (FTRL) with $R(p)$ as a regularizer. In fact, since universality results for mirror descent guarantee that some instantiation of FTRL achieves near-optimal rates for online linear optimization (as long as the action and loss sets are centrally convex) [SST11, GSJ24], this allows us to relate the performance of Theorem 3.1 directly to what rates are possible in online linear optimization.

**Corollary 1.2** (Informal restatement of Corollary C.6). *Let $\mathcal{P} \subseteq \mathbb{R}^d$ be a centrally symmetric convex set, and let $\mathcal{L} = \{y \in \mathbb{R}^d \mid \|y\|_* \leq 1\}$ for some norm $\|\cdot\|$. Then if there exists an algorithm for online linear optimization with action set $\mathcal{P}$ and loss set $\mathcal{L}$ that incurs regret at most $O(\sqrt{\rho T})$,* `TreeCal` *guarantees that the calibration error of its predictions is bounded by* $\mathsf{Cal}_T^{\|\cdot\|} \leq \epsilon T$ *for* $T \geq (\mathsf{diam}_{\|\cdot\|}(\mathcal{P})/\epsilon)^{O(\rho/\epsilon^2)}$.

Theorem 1.1 and its corollary allow us to immediately recover several existing and novel bounds on calibration error in a variety of settings:

- When $\mathcal{P}$ is the $d$-simplex $\Delta_d$ and $\|\cdot\|$ is the $\ell_1$-norm, the existence of the negative entropy regularizer $R(x) = \sum_{i=1}^{d} x_i \log x_i$ (which is 1-strongly convex w.r.t. the $\ell_1$ norm with range $\rho = \log d$) implies that the $\ell_1$ calibration error of `TreeCal` is at most $(1/\epsilon)^{O(\log d/\epsilon^2)} = d^{\tilde{O}(1/\epsilon^2)}$. This recovers the result of [Pen25].

- When $\mathcal{P}$ is the $\ell_2$ ball and $\|\cdot\|$ is the $\ell_2$ norm, the Euclidean regularizer ($R(x) = \|x\|^2$) implies a calibration bound of $(1/\epsilon)^{O(1/\epsilon^2)}$ (notably, this bound is independent of $d$).

It should be emphasized here that running `TreeCal` does not require any online linear optimization subroutine, nor any knowledge of these regularizers $R(x)$ or optimal rates $\rho$. `TreeCal` has no functional dependence on any specific $\|\cdot\|$. It achieves $\|\cdot\|$-calibration at the above rate (Theorem 1.1) for all $\|\cdot\|$ simultaneously. The `TreeCal` algorithm is nearly identical[5] to the algorithm of [Pen25] – both algorithms initialize a tree of sub-forecasters and at each round play a uniform combination of some subset of them (see Figure 1).

The novelty in our analysis stems from the observation that `TreeCal` is simply a specific instantiation of the `TreeSwap` swap regret minimization algorithm [DDFG24, PR24] and can be analyzed directly in this way. In particular, our analysis consists of the following steps:

1. First, minimizing calibration error can be reduced to minimizing swap regret, generalizing an idea of [LSS25, FKO+25]. That is, it is possible to assign the learner loss functions $\ell_t : \mathcal{P} \to \mathbb{R}$ at each round such that their calibration error is upper bounded by the gap between the total loss they received, and the minimal loss they could have received after applying an arbitrary "swap function" $\pi : \mathcal{P} \to \mathcal{P}$ to their predictions.

   In fact, any strongly convex function $R$ (w.r.t. the norm $\|\cdot\|$) gives rise to one such reduction, by setting the loss function $\ell_t(p)$ to equal the Bregman divergence $D_R(y_t|p)$.

2. Second, the `TreeSwap` algorithm of [DDFG24, PR24] provides a general recipe for converting external regret minimization algorithms into swap regret minimization algorithms. We obtain `TreeCal` by plugging in the Follow-The-Leader algorithm (the learning algorithm which simply always best responds to the current history) into `TreeSwap`.

3. Instead of analyzing the swap regret bound of `TreeSwap` with Follow-The-Leader (which may not have a good enough external regret bound, as discussed in Section 3.3), we instead analyze the swap regret of `TreeSwap` with *Be-The-Leader* (the fictitious algorithm that best responds to the current history, including the current round). Though it is not possible to actually implement Be-The-Leader due to its clairvoyance, we use it as a tool for analysis. We then relate the calibration error of `TreeSwap` with *Be-The-Leader* to that of `TreeSwap` with *Follow-The-Leader* using the fact that Be-The-Leader and Follow-The-Leader make similar predictions.

In the above step 1, we will choose $R$ to be $\|\cdot\|$-norm 1-strongly convex, which guarantees that $D_R(y|p) \geq \|y - p\|^2$. Going through the analysis, this actually leads to the stronger guarantee that `TreeCal` minimizes *squared-norm* calibration error.

**Theorem 1.3** (Informal restatement of Theorem 3.1). *Fix a convex set $\mathcal{P}$ and a norm $\|\cdot\|$. Assume there exists a function $R : \mathcal{P} \to \mathbb{R}$ that is 1-strongly-convex with respect to $\|\cdot\|$ and has range ($\max_{x \in \mathcal{P}} R(x) - \min_{p \in \mathcal{P}} R(x)$) at most $\rho$. Then `TreeCal` guarantees that the calibration error of its predictions is bounded by* $\mathsf{Cal}_T^{\|\cdot\|^2} \leq \epsilon T$ *for* $T \geq (\mathsf{diam}_{\|\cdot\|}(\mathcal{P})/\sqrt{\epsilon})^{O(\rho/\epsilon)}$.

Note here we have only singly-exponential dependence on $1/\epsilon$. We arrive at Theorem 1.1 as a corollary of this result by simply applying Cauchy-Schwarz. Finally, we strengthen the lower bound of [Pen25] by showing an exponential dependence on $1/\epsilon$ is necessary.

**Theorem 1.4** (Informal restatement of Theorem 4.3). *There is a sufficiently small constant $c > 0$ so that the following holds. Fix any $\epsilon > 0, d \in \mathbb{N}$. Then for any $T \leq \exp(c \cdot \min\{d^{1/14}, \epsilon^{-1/6}\})$, there is an oblivious adversary producing a sequence of outcomes so that any learning algorithm must incur $\ell_1$-calibration error* $\mathsf{Cal}_T^{\|\cdot\|_1} \geq \epsilon \cdot T$.

---

[5]One minor difference is that the algorithm of [Pen25] regularizes each sub-forecaster by slightly mixing their prediction with the uniform distribution, which `TreeCal` does not require.

Unlike the lower bound of [Pen25], this lower bound requires no specialized construction. Instead, it follows from the original observation of [FV98] that any algorithm for online calibration can be used to construct an algorithm for swap regret minimization by simply best responding to a sequence of calibrated predictions of the adversary's losses. The existing lower bound for swap regret in [DFG$^+$24] then immediately precludes the existence of sufficiently strong calibration bounds (e.g., of the form $d^{O(\log 1/\epsilon)}$, which was still allowed by the work of [Pen25]).

Using a similar technique, in Theorem D.2, we show a similar lower bound for $\ell_2$ calibration, namely that $\exp(\Omega(\min\{d^{1/14}, \epsilon^{-1/7}\}))$ time steps are needed to achieve $\ell_2$ calibration error at most $\epsilon \cdot T$. For $d \geq \epsilon^{-2}$, this bound is tight up a polynomial in the exponent.

We discuss additional related work in the appendix.

## 2 Setup

For a positive integer $n$, we let $[0 : n - 1]$ denote the sequence $0, 1, \ldots, n - 1$, and $[n]$ denote the sequence $1, 2, \ldots, n$. We say a convex set $\mathcal{S} \subseteq \mathbb{R}^d$ is *centrally symmetric* if $s \in \mathcal{S} \Leftrightarrow -s \in \mathcal{S}$ for all $s \in \mathbb{R}^d$. A norm $\|\cdot\|$ is a function corresponding to a convex, bounded, centrally-symmetric set $\mathcal{S}$ of the form $\|s\| = \inf \{c \in \mathbb{R}_{\geq 0} | s \in c\mathcal{S}\}$. The corresponding *dual norm* is defined $\|v\|_* = \sup \{\langle s, v \rangle \mid \|s\| \leq 1\}$.

### 2.1 Calibration

We consider the following setting of *multi-dimensional calibration*. Positive integers $d \in \mathbb{N}$ representing the number of dimensions and $T \in \mathbb{N}$ representing the number of rounds are given. We let $\mathcal{P} \subset \mathbb{R}^d$ denote a bounded convex subset of $\mathbb{R}^d$. An *adversary* and a *learning algorithm* interact for a total of $T$ timesteps; at each time step $t \in [T]$:

- The learning algorithm chooses a distribution[6] $\mathbf{x}_t \in \Delta(\mathcal{P})$ with finite support.
- The adversary observes $\mathbf{x}_t$ and chooses an *outcome* $y_t \in \mathcal{P}$.

In order for the learner to be calibrated, we would like the average outcome conditional on the learner making a specific prediction $p$ to be "close" to $p$. We formalize this as follows. For a point $p \in \mathcal{P}$, we define $\nu_p$ to be the average outcome conditioned on the learner predicting $p$, that is:

$$\nu_p := \frac{\sum_{t=1}^T \mathbf{x}_t(p) \cdot y_t}{\sum_{t=1}^T \mathbf{x}_t(p)}. \tag{1}$$

Fix a *distance measure* $D : \mathcal{P} \times \mathcal{P} \to \mathbb{R}_{\geq 0}$, namely an arbitrary non-negative valued function on $\mathcal{P} \times \mathcal{P}$. Given a distance measure $D$, we define the *D-calibration error* as follows:

$$\mathsf{Cal}_T^D(\mathbf{x}_{1:T}, y_{1:T}) := \sum_{p \in \mathcal{P}} \left(\sum_{t=1}^T \mathbf{x}_t(p)\right) \cdot D(\nu_p, p).$$

In the event that $D(p, q) = \|p - q\|$, we will write $\mathsf{Cal}_T^{\|\cdot\|}(\mathbf{x}_{1:T}, y_{1:T}) = \mathsf{Cal}_T^D(\mathbf{x}_{1:T}, y_{1:T})$, and we define $\mathsf{Cal}_T^{\|\cdot\|^2}(\mathbf{x}_{1:T}, y_{1:T})$ analogously.

### 2.2 Regret minimization

For a sequence of actions $p_1, \cdots, p_T \in \mathcal{P}$ and loss functions $\ell_1, \cdots, \ell_T : \mathcal{P} \to \mathbb{R}$, we define

$$\mathsf{ExtReg}_T(p_{1:T}, \ell_{1:T}) := \sup_{p^* \in \mathcal{P}} \sum_{t=1}^T \sum_{p \in \mathcal{P}} \ell_t(p_t) - \ell_t(p^*)$$

---

[6]Some authors refer to this setting as "pseudo-calibration" or "distributional calibration", and reserve the term "calibration" for the setting where the learner is required to randomly select a pure forecast $p_t \in \mathcal{P}$ each round instead of a distribution. In Appendix E we describe how to extend our results to this pure-strategy setting of calibration.

For a sequence of distributions $\mathbf{x}_1, \cdots, \mathbf{x}_T \in \Delta(\mathcal{P})$ and loss functions $\ell_1, \cdots, \ell_T : \mathcal{P} \to \mathbb{R}$, we define

$$\mathsf{FullSwapReg}_T(\mathbf{x}_{1:T}, \ell_{1:T}) := \sup_{\pi:\mathcal{P}\to\mathcal{P}} \sum_{t=1}^{T} \sum_{p\in\mathcal{P}} \mathbf{x}_t(p) \cdot (\ell_t(p) - \ell_t(\pi(p))). \tag{2}$$

Here, we adopt the convention of [FKO$^+$25], referring to the latter quantity as *Full* Swap Regret to emphasize that we consider *all* swap transformations $\pi : \mathcal{P} \to \mathcal{P}$ (instead of e.g. just linear transformations $\pi$).

Throughout, we consider the performance of *regret minimizing* algorithms. These algorithms sequentially map loss functions $\ell_1, \cdots, \ell_T$ to actions $p_1, \cdots, p_T$ or action distributions $\mathbf{x}_1, \cdots, \mathbf{x}_T$ with the goal of minimizing the above quantities. We consider the performance of these algorithms on adversarially selected loss functions from a set $\mathcal{L}$. Abusing notation slightly, for an external regret minimizing algorithm $\mathtt{Alg} : \mathcal{L}^T \to \mathcal{P}^T$, we define

$$\mathsf{ExtReg}_T(\mathtt{Alg}) := \sup_{\ell_{1:T}\in\mathcal{L}^T} \mathsf{ExtReg}_T(\mathtt{Alg}(\ell_{1:T}), \ell_{1:T}) \tag{3}$$

and for a full swap regret minimizing algorithm $\mathtt{Alg} : \mathcal{L}^T \to \Delta(\mathcal{P})^T$, we define

$$\mathsf{FullSwapReg}_T(\mathtt{Alg}) := \sup_{\ell_{1:T}\in\mathcal{L}^T} \mathsf{FullSwapReg}_T(\mathtt{Alg}(\ell_{1:T}), \ell_{1:T}).$$

We will denote the $t$th action played by $\mathtt{Alg}$ on a sequence of losses $\ell_{1:T}$ by $\mathtt{Alg}_t(\ell_{1:T})$. One important subclass of external regret minimization problems is the setting of *online linear optimization (OLO)*, where all loss functions in $\ell$ are linear. Here we slightly abuse notation and identify $\mathcal{L}$ with a subset of $\mathbb{R}^d$ (with the understanding that an element $\ell \in \mathcal{L}$ refers to the linear loss function $\ell(p) = \langle p, \ell \rangle$). Although we will never actually employ any OLO algorithms themselves, the calibration bounds we obtain will be closely related to optimal regret bounds for instances of OLO (we discuss this further in Section 2.4).

## 2.3   From swap regret to calibration

As noted in [LSS25, FKO$^+$25], calibration with a distance measure $D$ that corresponds to a *Bregman divergence* can be written as a full swap regret with loss functions given by the associated *proper scoring rule*. Given a convex function $R : \mathcal{P} \to \mathbb{R}$, the *Bregman divergence* associated to $R$, $D_R : \mathcal{P} \times \mathcal{P} \to \mathbb{R}_{\geq 0}$, is defined as[7]

$$D_R(y|p) := R(y) - R(p) - \langle \nabla R(p), y - p \rangle$$

Geometrically, this divergence is defined by taking the hyperplane tangent to $R$ at $p$ and computing the difference in height between $R$ and the hyperplane at $y$ (see Figure 2).

When viewed as a loss function in $p$, the Bregman divergence $D_R(y|p)$ also has the property that it is a *proper scoring rule*. This refers to the fact that if $y$ is drawn from some distribution $\mathbf{y} \in \Delta(\mathcal{P})$, the optimal response $p$ (to minimize the expected loss $D_R(y|p)$) is simply the expectation $\bar{y} = \mathbb{E}_{y\sim\mathbf{y}}[y]$. In particular, we have the following lemma.

**Lemma 2.1.** *For any* $\mathbf{y} \in \Delta(\mathcal{P})$ *and convex function* $R : \mathcal{P} \to \mathbb{R}$, *let* $\bar{y} = \mathbb{E}_{y\sim\mathbf{y}}[y]$. *and* $\overline{R(y)} = \mathbb{E}_{y\sim\mathbf{y}}[R(y)]$. *For all* $p \in \mathcal{P}$, $\mathbb{E}_{y\sim\mathbf{y}}[D_R(y|p)] = D_R(\bar{y}|p) + \overline{R(y)} - R(\bar{y})$. *In particular,* $\ell(p) = \mathbb{E}_{y\sim\mathbf{y}}[D_R(y|p)]$ *is minimized at* $p = \bar{y}$ *at a value of* $\overline{R(y)} - R(\bar{y})$ *(Figure 3).*

This implies the following connection between full swap regret and calibration.

**Lemma 2.2.** *Fix any convex function* $R : \mathcal{P} \to \mathbb{R}$. *For any sequence of distributions* $\mathbf{x}_1, \mathbf{x}_2, \ldots, \mathbf{x}_T \in \Delta(\mathcal{P})$ *and outcomes* $y_1, y_2, \ldots, y_T \in \mathcal{P}$, *define the sequence of loss functions* $\ell_1, \ell_2, \ldots, \ell_T$ *via* $\ell_t(p) = D_R(y_t|p)$. *Then,*

$$\mathsf{FullSwapReg}_T(\mathbf{x}_{1:T}, \ell_{1:T}) = \mathsf{Cal}_T^{D_R}(\mathbf{x}_{1:T}, y_{1:T}).$$

The proofs of Lemmas 2.1 and 2.2 may be found in Appendix B.

---

[7]In the event that $R$ is not differentiable, we can replace the $\nabla R(p)$ term with any element of the sub-gradient at $p$. When $\mathcal{P}$ is not open and $p$ is on the boundary, the $\nabla R(p)$ term represents the inward directional gradient.

## 2.4 Rates and regularization

In order to reduce our general calibration problem to a swap regret minimization problem (via Lemma 2.2), we will need to construct a convex function $R$ whose Bregman divergence upper bounds our distance measure. It turns out that the optimal choice of such a function is closely related to the design of optimal regularizers for online linear optimization. In this section, we describe this functional optimization problem and detail this connection.

We say that a convex function $R : \mathcal{P} \to \mathbb{R}$ is $\alpha$-*strongly convex* with respect to a given norm $\|\cdot\|$ if for any points $y, p \in \mathcal{P}$ it is the case that $R(y) \geq R(p) + \langle \nabla R(p), y - p \rangle + \alpha \|y - p\|^2$. Equivalently, the Bregman divergence must satisfy $D_R(y|p) \geq \alpha \|y - p\|^2$. Thus, $\|\cdot\|^2$-calibration error is bounded by $D_R$-calibration error if $R$ is $\|\cdot\|$-norm 1-strongly convex.

Our later analysis will need not only $R$ to be strongly convex with respect to our norm, but for the Bregman divergence to have a small maximal value. Motivated by this, we will say that a convex function $R : \mathcal{P} \to \mathbb{R}$ has *rate* $\rho$ with respect to a given norm $\|\cdot\|$ if: **(1)** $R$ is 1-strongly convex with respect to $\|\cdot\|$, and **(2)** the range of the Bregman divergence is at most $\rho$, i.e., $\max_{y,p \in \mathcal{P}} D_R(y|p) \leq \rho$. We define $\mathsf{Rate}(\mathcal{P}, \|\cdot\|)$ to be the infimum of the rates of all 1-strongly convex functions $R : \mathcal{P} \to \mathbb{R}$.

As mentioned earlier, we call this quantity a "rate" due to its connection with the optimal regret rates for online linear optimization. For a learning algorithm $\mathtt{Alg} : \mathcal{L}^T \to \mathcal{P}^T$, we defined (in (3)) $\mathsf{ExtReg}_T(\mathtt{Alg})$ to be the worst-case regret against any sequence $\ell_{1:T}$ of $T$ losses. It is known that for any fixed action set and loss set, the optimal worst-case regret bound is of the form $\sqrt{\mathsf{Rate}_{\mathsf{OLO}}(\mathcal{P}, \mathcal{L}) \cdot T} + o(\sqrt{T})$, for some constant $\mathsf{Rate}_{\mathsf{OLO}}(\mathcal{P}, \mathcal{L})$. Formally, we define $\mathsf{Rate}_{\mathsf{OLO}}(\mathcal{P}, \mathcal{L}) = \limsup_{T \to \infty} \inf_{\mathtt{Alg}} \frac{1}{T} \cdot \mathsf{ExtReg}_T(\mathtt{Alg})^2$.

One important class of learning algorithms for online linear optimization is the class of Follow-The-Regularized-Leader (FTRL) algorithms. Each algorithm in this class is specified by a convex "regularizer" function $R : \mathcal{P} \to \mathbb{R}$, and at round $t$ selects the action $p_t = \mathrm{argmin}_{p \in \mathcal{P}} \sum_{s=1}^{t-1} \langle p, \ell_t \rangle + R(p)$. The work of [SST11] and [GSJ24] shows that there always exists some instantiation of FTRL which achieves (up to a universal constant factor) the optimal regret rate of $\sqrt{\mathsf{Rate}_{\mathsf{OLO}}(\mathcal{P}, \mathcal{L}) \cdot T} + o(\sqrt{T})$ defined above. Moreover, the optimal regularizer for this instance can be constructed by solving a similar functional optimization problem over strongly convex regularizers $R$, as described in the following theorem.

**Theorem 2.3.** *Let $\mathcal{P}$ and $\mathcal{L}$ be centrally symmetric convex sets. Then, if the function $R : \mathcal{P} \to \mathbb{R}$ is 1-strongly-convex with respect to the norm $\|\cdot\|_{\mathcal{L}^*}$ and has range $\rho$ (i.e., $\max_{p \in \mathcal{P}} R(p) - \min_{p \in \mathcal{P}} R(p) = \rho$), then $\mathsf{Rate}_{\mathsf{OLO}}(\mathcal{P}, \mathcal{L}) \leq \rho$. Conversely, there exists a function $R : \mathcal{P} \to \mathbb{R}$ that is 1-strongly-convex with respect to $\|\cdot\|_{\mathcal{L}^*}$ and has range $O(\mathsf{Rate}_{\mathsf{OLO}}(\mathcal{P}, \mathcal{L}))$.*

*Proof.* The first result (that $\mathsf{Rate}_{\mathsf{OLO}}(\mathcal{P}, \mathcal{L}) \leq \rho$) follows from the standard analysis of FTRL – see e.g. Theorem 5.2 in [H+16]. The converse result follows from Theorem 2 of [GSJ24]. $\square$

Theorem 2.3 allows us to relate the quantity $\mathsf{Rate}(\mathcal{P}, \|\cdot\|)$ to the quantity $\mathsf{Rate}_{\mathsf{OLO}}(\mathcal{P}, \mathcal{L})$ (where $\mathcal{L}$ is chosen to be the unit dual norm ball). Note that there is a slight difference in the two functional optimization problems defined above – the one for $\mathsf{Rate}(\mathcal{P}, \|\cdot\|)$ asks us to bound the range of the Bregman divergence of $R$, while the one for $\mathsf{Rate}_{\mathsf{OLO}}(\mathcal{P}, \mathcal{L})$ asks us to bound the range of $R$ itself. While these two quantities do not directly bound each other (the negative entropy function $R(p) = \sum p_i \log p_i$ has bounded range over the simplex but unbounded Bregman divergence), we can nonetheless show that optimal solutions to one problem can be used to construct optimal solutions to the other problem of similar quality.

**Lemma 2.4.** *If the action set $\mathcal{P}$ is centrally symmetric and $\mathcal{L} = \{y \in \mathbb{R}^d \mid \|y\|_* \leq 1\}$ (i.e., the unit ball in the dual norm to $\|\cdot\|$), then $\mathsf{Rate}_{\mathsf{OLO}}(\mathcal{P}, \mathcal{L}) = \Theta(\mathsf{Rate}(\mathcal{P}, \|\cdot\|))$.*

## 3 Main result

We now describe our main algorithm for calibration, $\mathtt{TreeCal}$ (Algorithm 1). As we will see, it is equivalent to the $\mathtt{TreeSwap}$ algorithm for Full Swap Regret minimization ([DDFG24, PR24]; Algorithm 2), where the loss functions are given by appropriate Bregman divergences as determined by

Lemma 2.2. Moreover, `TreeCal` is effectively the same as the main algorithm of [Pen25]. However, the perspective that `TreeCal` can be viewed as a particular instance of `TreeSwap` (Lemma 3.2) is novel to this work, and it enables us to tackle a much more general set of calibration problems (Theorem 3.1). We first describe the `TreeCal` and `TreeSwap` algorithms, then state Theorem 3.1 which establishes our main upper bound for `TreeCal`, and finally discuss the proof of Theorem 3.1, which uses the `TreeSwap` algorithm as a tool in the analysis.

## 3.1 Algorithm description

Given some number of rounds $T \in \mathbb{N}$, `TreeCal` and `TreeSwap` sequentially produce distributions $\mathbf{x}_1, \cdots, \mathbf{x}_T \in \Delta(\mathcal{P})$. `TreeCal` receives from the adversary an outcome sequence $y_1, \cdots, y_T \in \mathcal{P}$ whereas `TreeSwap` receives loss functions $\ell_1, \cdots, \ell_T : \mathcal{P} \to \mathbb{R}$.

To describe how the algorithms use the adversary's actions to produce the distributions $\mathbf{x}_t$, we need some additional ntation. The algorithms take as input parameters $H, L \in \mathbb{N}$ satisfying $H \geq 2$ and $H^{L-1} \leq T \leq H^L$. We index time steps $t \in [T]$ via base-$H$ $L$-tuples: in particular, for $t \in [T]$, we let $t_1, \ldots, t_L \in [0 : H-1]$ be the base-$H$ representation of $t-1$; we will write $t-1 = (t_1 t_2 \cdots t_L)$. For all $0 \leq l \leq L$, for all $k \in [0 : H-1]^l$, let $\Gamma_k^{(l)} \subset [T]$ represent the interval of times $t$ with prefix $k$. That is, $t \in \Gamma_k^{(l)}$ iff $t_i = k_i$ for all $i \in [1 : l]$. These intervals may be arranged to form an $H$-ary depth-$L$ tree, where the children of $\Gamma_k^{(l)}$ are $\Gamma_{k0}^{(l+1)}, \Gamma_{k1}^{(l+1)}, \cdots, \Gamma_{k,H-1}^{(l+1)}$.[8]

Both `TreeCal` and `TreeSwap` operate by assigning an action $p_k^{(l)}$ to each node $\Gamma_k^{(l)}$ of the tree, except the root. At time $t$, both algorithms return the uniform distribution over the actions on the root-to-leaf-$t$ path, namely $\mathbf{x}_t := \mathrm{Unif}\left(\left\{p_{t_1}^{(1)}, p_{t_1 t_2}^{(2)}, \cdots, p_{t_1 t_2 \cdots t_L}^{(L)}\right\}\right)$ (see Figure 1). The algorithms differ in how the actions $p_k^{(l)}$ are chosen:

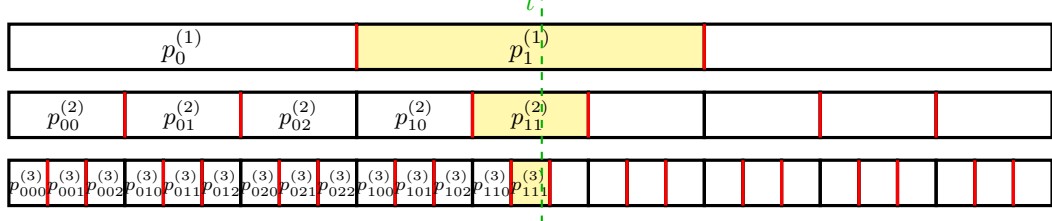

Figure 1: Visualization of the state of `TreeCal`/`TreeSwap` at time step $t$ (about half-way through the algorithm). For $H = 3$, we depict the intervals $\Gamma$ of the first three non-root levels of the tree ($l = 1, 2, 3$). Each rectangular node represents an interval, with sibling nodes separated by red lines. We represent the specific time step $t$ via the vertical dashed green line. The yellow intervals it intersects at each level correspond to the nodes on the root-to-leaf-$t$ path. Accordingly, $\mathbf{x}_t$ will be the uniform distribution over the labels $p$ of these yellow intervals. We see that the algorithm has committed to the labels of all intervals that started at or before time $t$, and has yet to label the future intervals.

- `TreeCal` (Algorithm 1) assigns actions to nodes as follows. For all $1 \leq l \leq L$, $k \in [0 : H-1]^{l-1}$, $h \in [0 : H-1]$, at the start of $\Gamma_{kh}^{(l)}$, `TreeCal` sets $p_{kh}^{(l)}$ to be the average over all $y_t$ that have been observed thus far in the parent interval $\Gamma_k^{(l-1)}$. That is,

$$p_{kh}^{(l)} = \frac{1}{hH^{L-l}} \sum_{i=0}^{h-1} \sum_{t \in \Gamma_{ki}^{(l)}} y_t \qquad (4)$$

- The more general `TreeSwap` algorithm (Algorithm 2) also takes as a parameter an external regret-minimizing algorithm `Alg`, which operates with horizon of length $H$: we denote the resulting algorithm by `TreeSwap.Alg`. `TreeSwap.Alg` associates each internal node of the tree, $\Gamma_k^{(l-1)}$ (with $1 \leq l \leq L$), with an instance `Alg`, denoted $\mathrm{Alg}_k^{(l-1)}$. The subroutine $\mathrm{Alg}_k^{(l-1)}$ is responsible for choosing the actions $p_{k0}^{(l)}, p_{k1}^{(l)}, \cdots, p_{k(H-1)}^{(l)}$. It does so by responding to the average losses over each of its child intervals. In particular: at the end of

---

[8]We ignore the truncated branches that exist if $T < H^L$.

each child interval $\Gamma_{kh}^{(l)}$, we pass $\texttt{Alg}_k^{(l-1)}$ the average loss over that interval. $\texttt{Alg}_k^{(l-1)}$ then outputs the action $p_{k(h+1)}^{(l)}$ assigned to the next child interval.

## 3.2 Main result

Theorem 3.1 upper bounds the calibration error of $\texttt{TreeCal}$ with respect to the squared norm $\|\cdot\|^2$.

**Theorem 3.1** (Main theorem). *Let $\mathcal{P} \subset \mathbb{R}^d$ be a bounded convex set and $\|\cdot\|$ be an arbitrary norm. Then, $\texttt{TreeCal}$ (Algorithm 1) guarantees that for an arbitrary sequence of outcomes $y_1, \ldots, y_T \in \mathcal{P}$, the $\|\cdot\|^2$ calibration error of its predictions $\mathbf{x}_1, \ldots, \mathbf{x}_T \in \Delta(\mathcal{P})$ is bounded as follows:*

$$\mathsf{Cal}_T^{\|\cdot\|^2}(\mathbf{x}_{1:T}, y_{1:T}) \leq \epsilon T \quad \text{for} \quad T \geq (\mathsf{diam}(\mathcal{P})/\sqrt{\epsilon})^{O(\mathsf{Rate}(\mathcal{P}, \|\cdot\|)/\epsilon)}$$

It is straightforward to derive from Theorem 3.1 via an application of Jensen's inequality an upper bound on the calibration error of $\texttt{TreeCal}$ with respect to the (non-squared) norm $\|\cdot\|$, as stated in Theorem 1.1; see Corollary C.5. In Appendix E, we additionally consider a variant of $\texttt{TreeCal}$ which plays *pure actions* in $\mathcal{P}$ (i.e., not distributions) by sampling from the distributions $\mathbf{x}_t$ for each $t \in [T]$. We show that the *pure calibration* error of this variant can be bounded by a similar quantity to that in Theorem 3.1.

## 3.3 Outline of the proof of Theorem 3.1

**Step 1: Reduction from calibration error to swap regret.** Let us choose a convex function $R : \mathcal{P} \to \mathbb{R}$ given $\mathcal{P}, \|\cdot\|$ as described in Section 2.4. The first step in the proof of Theorem 3.1 is to reduce the problem of minimizing (squared-norm) calibration error to that of minimizing full swap regret for an appropriate sequence of loss functions. In particular, for any sequence $\mathbf{x}_1, \ldots, \mathbf{x}_T \in \Delta(\mathcal{P})$ and $y_1, \ldots, y_T \in \mathcal{P}$, we have

$$\mathsf{Cal}_T^{\|\cdot\|^2}(\mathbf{x}_{1:T}, y_{1:T}) \leq \mathsf{Cal}_T^{D_R}(\mathbf{x}_{1:T}, y_{1:T}) = \mathsf{FullSwapReg}_R(\mathbf{x}_{1:T}, \ell_{1:T}), \tag{5}$$

where $\ell_t : \mathcal{P} \to \mathbb{R}$ is the loss function given by $\ell_t(p) := D_R(y_t|p)$: the inequality uses strong convexity of $R$, and the subsequent equality uses Lemma 2.2.

**Step 2: Equivalence with $\texttt{TreeSwap}$.** Thus, it suffices to find an algorithm which minimizies the full swap regret quantity on the right-hand side of (5). Fortunately, the $\texttt{TreeSwap}$ algorithm is known to do exactly this! (See Theorem C.1, from [DDFG24], for a formal statement for the swap regret bound of $\texttt{TreeSwap}$.) In order to apply the swap regret bound of Theorem C.1, we need to ensure that the $\texttt{TreeCal}$ algorithm is an instantiation of $\texttt{TreeSwap.Alg}$ for an appropriate choice of (a) the loss functions fed as input to $\texttt{TreeSwap}$ and (b) the $\texttt{Alg}$ subroutine. The loss functions have already been defined: given a sequence $y_1, \ldots, y_T$, recall that we chose $\ell_t(p) := D_R(y_t|p)$. Moreover, we let the $\texttt{Alg}$ subroutine be given by *Follow-the-Leader* (FTL), which simply chooses an action at each step minimizing the sum of losses up to the previous time step. The following lemma shows that $\texttt{TreeSwap}$ with the losses $\ell_t$ and the FTL subroutine produces the same action distributions as $\texttt{TreeCal}$:

**Lemma 3.2.** *Let $\mathcal{P} \subset \mathbb{R}^d$ be a bounded convex set and let $R : \mathcal{P} \to \mathbb{R}$ be a convex function. For a sequence of loss functions $\ell_1, \cdots, \ell_H : \mathcal{P} \to \mathbb{R}$, define $\texttt{FTL}_h(\ell_{1:H}) = \arg\min_{p \in \mathcal{P}} \sum_{s=1}^{h-1} \ell_s(p)$. For all sequences of outcomes $y_{1:T} \in \mathcal{P}^T$, the action distributions $\mathbf{x}_t$ produced by $\texttt{TreeCal}$ on $y_{1:T}$ equal those produced by $\texttt{TreeSwap.FTL}$ on loss functions $\ell_t(p) = D_R(y_t|p)$ for all $t$.*

The proof of Lemma 3.2 (given in full in the appendix) is a straightforward consequence of the fact that the Bregman divergence is a proper scoring rule: the action $p \in \mathcal{P}$ minimizing an average of Bregman divergences $D_R(y|p)$ is simply the average of the constituent points $y$ (Lemma 2.1).

**Step 3: Applying the swap regret bound of $\texttt{TreeSwap}$ to BTL.** Finally, we want to apply the main result of [DDFG24] (restated as Theorem C.1) to bound the full swap regret for the iterates $\mathbf{x}_{1:T}$ produced by $\texttt{TreeSwap.Alg}$, for an appropriate choice of $\texttt{Alg}$. The most natural way to do so would be to try to directly apply this result in the case when $\texttt{Alg} = \texttt{FTL}$ (which corresponds to how we actually implement $\texttt{TreeSwap}$). However, applying this theorem requires an external regret bound on FTL for an arbitrary sequence of losses. While FTL is known to possess strong external regret

bounds in some situations (e.g., when all the loss functions are strongly convex), the loss functions $p \mapsto D_R(y|p)$ are not necessarily even convex in $p$ and so it is not a priori clear how to establish such bounds.

Instead, the main idea is to consider the "Be-The-Leader" algorithm BTL, which is the same as FTL but where actions are shifted ahead in time by 1 time step: in particular, the action chosen by BTL at time step $h$ given a sequence $\ell_1, \ell_2, \ldots, \ell_H : \mathcal{P} \to \mathbb{R}$ is $\mathtt{BTL}_h(\ell_{1:H}) = \mathtt{FTL}_{h+1}(\ell_{1:H}) = \operatorname{argmin}_{p \in \mathcal{P}} \sum_{s=1}^{h} \ell_s(p)$. BTL is not implementable since its action at time step $h$ depends on the (unobserved) loss $\ell_h$ at that time step. However, since its regret is always non-positive (i.e., $\mathsf{ExtReg}_H(\mathtt{BTL}) \leq 0$ for any $H$), if we apply Theorem C.1 to the algorithm $\mathtt{TreeSwap.BTL}$, we get that $\mathsf{FullSwapReg}_T(\mathtt{TreeSwap.BTL}) \leq \epsilon \cdot T$ as long as $T \geq H^{O(\rho/\epsilon)}$ for *any* choice of $H$ (the arity parameter $H$ used in $\mathtt{TreeSwap}$). Using (5), this implies that the *calibration error* of the iterates produced by $\mathtt{TreeSwap.BTL}$ can also be bounded above by $\epsilon \cdot T$.

Of course, this result on its own is uninteresting (since BTL is unimplementable, as mentioned above). However, the key insight is that we can show that the actions chosen by $\mathtt{TreeSwap.BTL}$ are close to (as measured by the norm $\|\cdot\|$) those chosen by $\mathtt{TreeSwap.FTL}$, which in turn is equivalent to $\mathtt{TreeCal}$ (Lemma 3.2). This closeness is an immediate consequence of the fact that the actions chosen by FTL for our loss functions $D_R(y_1|\cdot), D_R(y_2|\cdot), \ldots$ are simply the empirical average of all actions $y_1, y_2, \ldots \in \mathcal{P}$ of the adversary up to the previous time step.[9] In turn, we can use this closeness to show that the calibration error of $\mathtt{TreeSwap.FTL}$ is close to that of $\mathtt{TreeSwap.BTL}$. This latter part of the argument becomes slightly tricky due to the possibility that different nodes of the tree might output the same action $p \in \mathcal{P}$; accordingly, we need to work with a *labeled* variant of the action set and bound the swap regret over this labeled variant; see Appendix C for further details.

# 4   Lower bound

To prove our calibration lower bound, we make use of the following swap regret lower bound.

**Theorem 4.1** (Theorem 4.1 of [DFG$^+$24])**.** *There is a sufficiently small constant $c_{4.1} > 0$ so that the following holds. Fix any $\epsilon > 0$. For any $d \in \mathbb{N}$, there is a subset $\mathcal{X} \subset [-1, 1]^d$ so that the following holds for any $T \leq \exp\left(c_{4.1} \min\{d^{1/14}, \epsilon^{-1/6}\}\right)$. There is an oblivious adversary producing a sequence $v_1, \ldots, v_T$ with $\|v_t\|_1 \leq 1$ and $\|v_t\|_\infty \leq \max\{d^{-13/14}, \epsilon^{13/6}\}$ for all $t$, which satisfies the following property. For linear loss functions $\ell(x, v) = \langle v, x \rangle$ for vectors $v \in \mathbb{R}^d$ and $x \in \mathbb{R}^d$, any learning algorithm producing $\mathbf{x}_1, \ldots, \mathbf{x}_T \in \Delta(\mathcal{X})$,*

$$\mathsf{FullSwapReg}_T(\mathbf{x}_{1:T}, \ell(\cdot, v_{1:T})) = \sup_{\pi:\mathcal{X}\to\mathcal{X}} \sum_{t=1}^{T} \sum_{p \in \mathcal{X}} \mathbf{x}_t(p) \cdot (\langle v_t, p \rangle - \langle v_t, \pi(p) \rangle) \geq \epsilon \cdot T.$$

We leverage the classic reduction from swap-regret minimization to calibration [FV98]: by producing calibrated predictions of the upcoming loss and best-responding to it, we can effectively minimize swap regret. This is formalized in the following lemma, proved in Appendix D.

**Lemma 4.2.** *Fix a set $\mathcal{P} \subset \mathbb{R}^d$, a norm $\|\cdot\|$, and write $D(p, p') := \|p - p'\|$. Suppose that, for some $\epsilon > 0, T \in \mathbb{N}$, there is an algorithm which chooses $\mathbf{x}_1, \ldots, \mathbf{x}_T \in \Delta(\mathcal{P})$ and which ensures that for every oblivious adversary choosing $y_1, \ldots, y_T \in \mathcal{P}$, we have $\mathsf{Cal}_T^D(\mathbf{x}_{1:T}, y_{1:T}) \leq \epsilon \cdot T$. Then for every set $\mathcal{P}' \subset \mathbb{R}^d$, there is an algorithm which chooses $\mathbf{x}'_1, \ldots, \mathbf{x}'_T \in \Delta(\mathcal{P}')$ and which ensures that for every oblivious adversary choosing $y_1, \ldots, y_T \in \mathcal{P}$, we have*

$$\mathsf{FullSwapReg}_T(\mathbf{x}'_{1:T}, \ell(\cdot, y_{1:T})) \leq \epsilon \cdot T \cdot \mathsf{diam}_{\|\cdot\|_\star}(\mathcal{P}').$$

Combining these two ideas, we demonstrate that an algorithm $\epsilon$-calibrated predictions of outcomes on the simplex in $T \leq \exp(\mathsf{poly}(1/\epsilon))$ rounds could be used in Lemma 4.2 to achieve a swap regret algorithm contradicting Theorem 4.1. This gives the following (proved in Appendix D).

---

[9] An observant reader might note that this same argument also lets us provide bounds on the regret of FTL for these losses. One subtlety in the analysis is that we obtain better calibration bounds by bounding the distance between the predictions of FTL and BTL in the $\|\cdot\|$ norm rather than in the losses $D_R(y_t|\cdot)$, and so it is important that we directly analyze $\mathtt{TreeSwap.BTL}$ instead of $\mathtt{TreeSwap.FTL}$ (the latter causes us to pick up an extra factor related to the *smoothness* of $R$).

**Theorem 4.3.** *There is a sufficiently small constant $c > 0$ so that the following holds. Write $D(p, p') = \|p - p'\|_1$, and fix any $\epsilon > 0, d \in \mathbb{N}$. Then for any $T \leq \exp(c \cdot \min\{d^{1/14}, \epsilon^{-1/6}\})$, there is an oblivious adversary producing a sequence $y_1, \ldots, y_T \in \Delta^d$ so that for any learning algorithm producing $\mathbf{x}_1, \ldots, \mathbf{x}_T \in \Delta(\Delta^d)$, $\mathsf{Cal}_T^D(\mathbf{x}_{1:T}, y_{1:T}) \geq \epsilon \cdot T$.*

In Theorem D.2 (see Appendix D.2), we show a similar lower bound for $\ell_2$ calibration over the unit $\ell_2$ ball.

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

# A  Additional Related Work

There is a large range of other existing work on online (sequential) calibration [Daw82, FV97, FV98, QV21, DDF⁺24, Har22, Fos99, FL99, KF08, MSA07, MS10, AM11, HK12, FH18, LSS24, NRRX23, KLST23, GJRR24, QZ24, ACRS25]. We briefly survey some of these areas below.

**Binary outcomes.**   For binary outcomes (i.e., one-dimensional calibration), classical results of [FV97, Fos99, BM07, AM11] demonstrate that it is possible to efficiently guarantee $O(T^{2/3})$ $\ell_1$-calibration. The optimal possible rates for $\ell_1$-calibration remain a major unsolved problem in online learning. Recently [QV21] improved over the naive lower bound of $\Omega(\sqrt{T})$ by demonstrating a lower bound of $\Omega(T^{0.528})$; this was further improved to $\Omega(T^{0.543})$ by [DDF⁺24], who also improved on the upper bound, demonstrating the existence of an algorithm with $O(T^{2/3-\epsilon})$ calibration for some constant $\epsilon > 0$.

**Calibration and swap regret.**   The connection between calibration and swap regret has been acknowledged since the earliest works on swap regret. For example, the earliest algorithms for minimizing swap regret worked by best responding to online calibrated predictions [FV97] (later algorithms for swap regret minimization, such as [BM07] and [DDF⁺24] obtain better swap regret bounds by side-stepping the need to generate calibrated predictions). In the other direction, several works minimize calibration via relating it to a swap regret that can then be minimized [FKO⁺25, LSS25, AM11, Fos99].

**Other forms of calibration.**   Due to the difficulty of minimizing (high-dimensional) calibration, there has been a line of work on designing forecasting algorithms that minimize weaker forms of calibration that recover some of the important guarantees of calibration (e.g., trustworthy-ness by a decision-maker). These include *distance from calibration* [BGHN23, QZ24, ACRS25], *omni-prediction error / U-calibration* [KLST23, LSS24, GJRR24], *calibration conditioned on downstream outcomes* [NRRX23], and *prediction for downstream swap regret* [RS24, HW24]. Other work focuses on minimizing notions of calibration designed to lead to specific classes of equilibria, e.g. weak calibration [HK12], deterministic calibration [KF08], and smooth calibration [FH18].

# B  Proofs of preliminary results

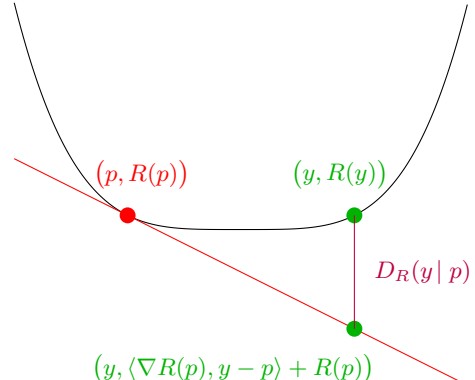

Figure 2: Geometric depiction of the Bregman divergence from $p$ to $y$.

*Proof of Lemma 2.1.*

$$\mathbb{E}_{y\sim\mathbf{y}}[D_R(y|p)] = \mathbb{E}_{y\sim\mathbf{y}}\left[R(y) - R(p) - \langle\nabla R(p), y - p\rangle\right]$$
$$= \overline{R(y)} - R(p) - \langle\nabla R(p), \bar{y} - p\rangle$$
$$= D_R(\bar{y}|p) + \overline{R(y)} - R(\bar{y})$$

See Figure 3 for a visual proof.  □

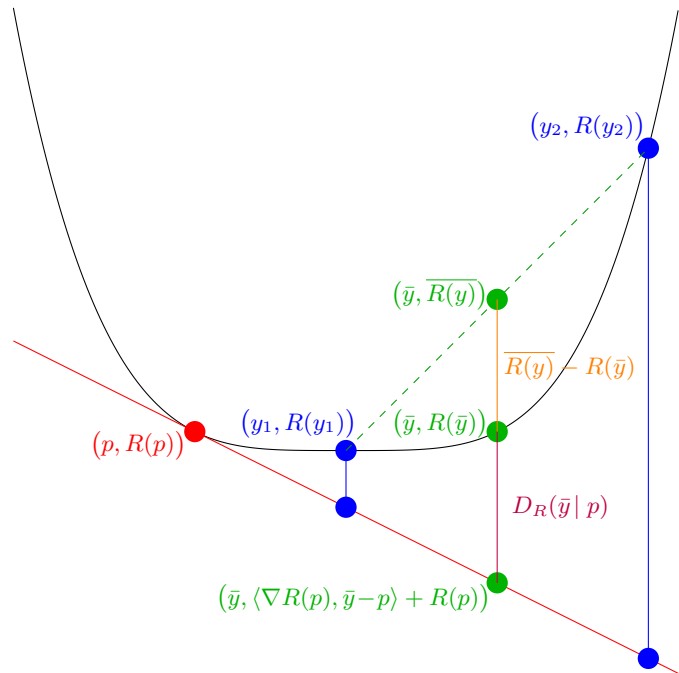

Figure 3: [Proof of Lemma 2.1] the average Bregman divergence (orange + purple) decomposes into the Jensen error (orange) and the Bregman divergence to the mean (purple). For example, when $R(p) = \|p\|_2^2$, $D_R(y|p) = \|y - p\|_2^2$ and we recover the bias-variance decomposition.

*Proof of Lemma 2.2.* Fix any $p \in \mathcal{P}$, and consider the quantity $\max_{p^* \in \mathcal{P}} \sum_t \mathbf{x}_t(p)(D_R(y_t|p) - D_R(y_t|p^*))$. By considering the distribution $\mathbf{y}$ that has weight $\mathbf{x}_t(p)/\sum_t \mathbf{x}_t(p)$ on $y_t$, Lemma 2.1 implies that this quantity is maximized when $p^* = \nu_p = (\sum_t \mathbf{x}_t(p)y_t)/(\sum_t \mathbf{x}_t(p))$. At this optimal value of $p^*$, this quantity can be rewritten as:

$$
\begin{aligned}
&\sum_t \mathbf{x}_t(p)(D_R(y_t|p) - D_R(y_t|\nu_p)) \\
=\; &\sum_t \mathbf{x}_t(p)\left[(R(y_t) - R(p) - \langle \nabla R(p), y_t - p\rangle) - (R(y_t) - R(\nu_p) - \langle \nabla R(\nu_p), y_t - \nu_p\rangle)\right] \\
=\; &\sum_t \mathbf{x}_t(p)\left[(R(\nu_p) - R(p) - \langle \nabla R(p), \nu_p - p\rangle) + \langle \nabla R(\nu_p) - \nabla R(p), y_t - \nu_p\rangle\right] \\
=\; &\sum_t \mathbf{x}_t(p)D_R(\nu_p|p) + \left\langle \nabla R(\nu_p) - \nabla R(p), \sum_t \mathbf{x}_t(p)(y_t - \nu_p)\right\rangle \\
=\; &\sum_t \mathbf{x}_t(p)D_R(\nu_p|p).
\end{aligned}
$$

(Here the last term vanishes since $\sum_t \mathbf{x}_t(p)y_t = \sum_t \mathbf{x}_t(p)\nu_p$). We therefore have that:

$$
\begin{aligned}
\mathsf{FullSwapReg}_T(\mathbf{x}_{1:T}, \ell_{1:T}) &= \sup_{\pi:\mathcal{P}\to\mathcal{P}} \sum_{t=1}^{T} \sum_{p\in\mathcal{P}} \mathbf{x}_t(p)\cdot(\ell_t(p) - \ell_t(\pi(p))) \\
&= \sum_{p\in\mathcal{P}} \max_{p^*\in\mathcal{P}} \sum_{t=1}^{T} \mathbf{x}_t(p)\cdot(\ell_t(p) - \ell_t(p^*)) \\
&= \sum_{p\in\mathcal{P}} \max_{p^*\in\mathcal{P}} \sum_{t=1}^{T} \mathbf{x}_t(p)\cdot(D_R(y_t|p) - D_R(y_t|\nu_p)) \\
&= \sum_{p\in\mathcal{P}} \sum_{t} \mathbf{x}_t(p) D_R(\nu_p|p) \\
&= \mathsf{Cal}_T^{D_R}(\mathbf{x}_{1:T}, y_{1:T}).
\end{aligned}
$$

$\square$

*Proof of Lemma 2.4.* Note that if we define $\mathcal{L} = \{y \in \mathbb{R}^d \mid \|y\|_* \le 1\}$ to be the unit dual norm ball for some norm $\|\cdot\|$, then by duality the norm $\|\cdot\|_{\mathcal{L}^*}$ corresponding to $\mathcal{L}^*$ is simply the original norm $\|\cdot\|$. It therefore suffices to show that given a 1-strongly convex function $R$ with bounded range $\rho$, it is possible to construct a 1-strongly convex function $R'$ with bounded Bregman divergence $O(\rho)$ (and vice versa).

Assume $R(p)$ is 1-strongly convex and satisfies $\max_{p\in\mathcal{P}} R(p) - \min_{p\in\mathcal{P}} R(p) = \rho$. Define $R'(p) = 4R\left(\frac{p}{2}\right)$ (since $\mathcal{P}$ is centrally symmetric, $p/2$ is guaranteed to belong to $\mathcal{P}$). If $R$ is 1-strongly convex, then $R(p/2)$ is $1/4$-strongly convex, and so $R'(p)$ is also 1-strongly convex. We claim the maximum Bregman divergence of $R'$ is at most $O(\rho)$. To show this, we first argue that for any $z_1, z_2 \in \mathcal{P}$, $\left\langle \nabla R(\frac{z_1}{2}), z_2 \right\rangle \le 2\rho$. To see this, note that since $R(p)$ is convex and has range bounded by $\rho$, we have that $\rho \ge R(p) - R(\frac{z_1}{2}) \ge \left\langle \nabla R(\frac{z_1}{2}), x - \frac{z_1}{2} \right\rangle$. If we set $p = \frac{z_1+z_2}{2}$, it then follows that $\left\langle \nabla R(\frac{z_1}{2}), z_2 \right\rangle \le 2\rho$. Now, note that

$$
\begin{aligned}
\max_{y,p\in\mathcal{P}} D_{R'}(y|p) &= R'(y) - R'(p) - \langle \nabla R'(p), y - p \rangle \\
&= R\left(\frac{y}{2}\right) - R\left(\frac{p}{2}\right) - \frac{1}{2}\left\langle \nabla R\left(\frac{p}{2}\right), y - p \right\rangle \\
&\le \left| R\left(\frac{y}{2}\right) - R\left(\frac{p}{2}\right) \right| + \left\langle \nabla R\left(\frac{p}{2}\right), \frac{y-p}{2} \right\rangle \le 3\rho.
\end{aligned}
$$

Conversely, if $R(p)$ is 1-strongly convex and satisfies $\max_{y,p\in\mathcal{P}} D_R(y|p) \le \rho$, define $R'(p) = R(p) - \langle \nabla R(0), p \rangle - R(0)$ (i.e., subtracting a linear function to make zero a minimizer of $R'(p)$). Since $R$ and $R'$ differ by a linear function, $R'$ is also 1-strongly convex. But also, note that $D_R(y|0) = R(y) - R(0) - \langle \nabla R(0), y \rangle = R'(y)$; since $D_R$ is bounded in range by $\rho$, it follows that so is $R'$.

$\square$

## C  Proof of Theorem 3.1

In this section, we prove Theorem 3.1. First, in Appendix C.1, we introduce a slightly stronger notion of calibration error and swap regret to deal with a technicality in the proof. We then give the proof of Theorem 3.1.

### C.1  Labeled calibration and swap regret

**Intuition.** Recall that the `TreeCal` algorithm labels each interval $\Gamma_k^{(l)}$ of the tree with some action, $p_k^{(l)} \in \mathcal{P}$. At each time step $t$, the algorithm outputs the uniform distribution over all $p_k^{(l)}$

---
**Algorithm 1** TreeCal$(\mathcal{P}, T, H, L)$
---
**Require:** Action set $\mathcal{P} \subset \mathbb{R}^d$, time horizon $T$, parameters $H, L$ with $T \leq H^L$.
 1: **for** $1 \leq t \leq T$ **do**
 2:     Write the base-$H$ representation of $t - 1$ as $t = (h_1 \cdots h_L)$, for $h_1, \ldots, h_L \in [0 : H - 1]$.
 3:     **for** $1 \leq l \leq L$ **do**
 4:         Write $\bar{k} := (h_1 \cdots h_{l-1}) \in [0 : H - 1]^{l-1}$.
 5:         **if** $h_{l+1} = \cdots = h_L = 0$ or $l = L$ **then**
 6:             If $h_l > 0$, define $\nu_{k,h_l-1}^{(l)} := \frac{1}{H^{L-l}} \cdot \sum_{s \in \Gamma_{k,h_l-1}^{(l)}} y_s$.
 7:             Define $p_{k,h_l}^{(l)} := \frac{1}{h_l} \sum_{i=0}^{h_l-1} \nu_{k,i}^{(l)}$ if $h_l > 0$, otherwise choose arbitrary $p_{k,h_l}^{(l)} \in \mathcal{P}$.
 8:         **end if**
 9:     **end for**
10:     Output the uniform mixture $\mathbf{x}_t := \mathrm{Unif}(\{p_{h_1}^{(1)}, \ldots, p_{h_1 \cdots h_L}^{(L)}\})$, and observe $y_t$.
11: **end for**
---

with $\Gamma_k^{(l)} \ni t$. When evaluating the calibration error, suppose that the actions $p_k^{(l)}$ are all distinct, for $l \in [L], k \in [0 : H - 1]^{l-1}$ (as we discuss below, this case is in some sense the "worst case"). In this event, each action $p_k^{(l)}$ is compared to the average outcome over the interval $\Gamma_k^{(l)}$: $\bar{y}_k^{(l)} = \frac{1}{|\Gamma_k^{(l)}|} \sum_{t \in \Gamma_k^{(l)}} y_t$. Formally, this would give

$$\mathsf{Cal}_T^D(\mathbf{x}_{1:T}, y_{1:T}) = \sum_{l=1}^{L} \frac{H^{L-l}}{L} \sum_{k \in [H]^l} D\left(\bar{y}_k^{(l)}, p_k^{(l)}\right). \tag{6}$$

as each level $l$ action is selected with $\frac{1}{L}$ mass for $H^{L-l}$ rounds.

If it happened that two distinct intervals $\Gamma_{k_1}^{(l_1)}, \Gamma_{k_2}^{(l_2)}$ were assigned the same action $p = p_{k_1}^{(l_1)} = p_{k_1}^{(l_1)}$, then the calibration error would be *at most* the quantity on the right-hand side of (6) (by Jensen's inequality). In particular, rather than having to compare $p$ to two potentially distinct quantities $D(\bar{y}_{k_1}^{(l_1)}, p), D(\bar{y}_{k_2}^{(l_2)}, p)$, the mass placed on $p$ would be categorized under the same forecast and we would only compare $p$ to an appropriately-weighted average of $\bar{y}_{k_1}^{(l_1)}$ and $\bar{y}_{k_2}^{(l_2)}$.

For technical reasons, it will turn out to be necessary to upper bound the "worst case quantity" on the right-hand side of (6) (and an analogous version for swap regret), even in the even that the actions $p_k^{(l)}$ are *not all distinct*. To streamline our notation, we introduce a generalization of these quantities which apply for arbitrary algorithms, which we call *labeled* calibration error and *labeled* swap regret.

**Formal definitions.** Given a convex set $\mathcal{P} \subset \mathbb{R}^d$, we define its *labeled* extension to be $\bar{\mathcal{P}} := \mathcal{P} \times \{0, 1\}^\star$, i.e., elements of $\bar{\mathcal{P}}$ are tuples $(p, \sigma)$, where $\sigma \in \{0, 1\}^\star$ is a string that is said to *label* $p$. For a loss function $\ell : \mathcal{P} \to \mathbb{R}$, we extend its domain to $\bar{\mathcal{P}}$ in the natural way, i.e., $\ell((p, \sigma)) := \ell(p)$ for $(p, \sigma) \in \bar{\mathcal{P}}$. Given a sequence of distributions over the labeled extension, $\mathbf{x}_1, \ldots, \mathbf{x}_T \in \Delta(\bar{\mathcal{P}})$, and loss functions $\ell_1, \ldots, \ell_T : \mathcal{P} \to \mathbb{R}$, we define

$$\mathsf{FullSwapReg}_T(\mathbf{x}_{1:T}, \ell_{1:T}) := \sup_{\pi : \bar{\mathcal{P}} \to \bar{\mathcal{P}}} \sum_{t=1}^{T} \sum_{p \in \bar{\mathcal{P}}} \mathbf{x}_t(p) \cdot (\ell_t(p) - \ell_t(\pi(p))).$$

In words, the full swap regret of $\mathbf{x}_{1:T}$ with respect to $\ell_{1:T}$ is defined identically as in (2) except that the swap function $\pi$ can now depend on the label $\sigma$. In particular, the labeled extension allows us to consider a more refined notion of swap regret where identical actions played in different rounds can be swapped (via $\pi$) to different alternatives as long as they have different labels.

In a similar manner we define the calibration error for a sequence of labeled distributions: given $\mathbf{x}_1, \ldots, \mathbf{x}_T \in \Delta(\bar{\mathcal{P}})$ and $y_1, \ldots, y_T \in \mathcal{P}$, we define

$$\mathsf{Cal}_T^D := \sum_{(p,\sigma) \in \bar{\mathcal{P}}} \left(\sum_{t=1}^{T} \mathbf{x}_t((p, \sigma))\right) \cdot D(\nu_{(p,\sigma)}, p), \qquad \nu_{(p,\sigma)} := \frac{\sum_{t=1}^{T} \mathbf{x}_t((p, \sigma)) \cdot y_t}{\sum_{t=1}^{T} \mathbf{x}_t((p, \sigma))}.$$

The main result of [DDFG24] shows that the swap regret of `TreeSwap` is bounded, even when one labels the action produced at each node of the tree by the node of the tree. This labeled variant of `TreeSwap` is given in Algorithm 2. It functions exactly as discussed in Section 3.1, except that the distribution $\mathbf{x}_t$ output at time step $t$ is in $\Delta(\bar{\mathcal{P}})$ instead of $\Delta(\mathcal{P})$. In particular, each $p_k^{(l)} \in \mathcal{P}$ in the support of $\mathbf{x}_t$ is labeled by the tuple $k \in [0 : H-1]^l$.[10]

**Theorem C.1** (`TreeSwap`; Theorem 3.1 of [DDFG24]). *Suppose that $H, L \in \mathbb{N}$ satisfy $H \geq 2$ and $H^{L-1} \leq T \leq H^L$. For bounded convex action set $\mathcal{P} \subset \mathbb{R}^d$ and loss function set $\mathcal{L} \subset \{\ell : \mathcal{P} \to [0, b]\}$, let $\mathtt{Alg}_H : \mathcal{L}^H \to \mathcal{P}^H$ be any algorithm. Then, the labeled `TreeSwap` algorithm (Algorithm 2) parametrized by $T, H, L, \mathcal{P}, \mathcal{L}, \mathtt{Alg}_H$ outputs labeled distributions $\mathbf{x}_1, \ldots, \mathbf{x}_T \in \Delta(\bar{\mathcal{P}})$ satisfying the following: for any sequence $\ell_1, \ldots, \ell_T \in \mathcal{L}$,*

$$\mathsf{FullSwapReg}_T(\mathbf{x}_{1:T}, \ell_{1:T}) \leq T \cdot \left( \frac{\mathsf{ExtReg}_H(\mathtt{Alg}_H)}{H} + \frac{3b}{L} \right).$$

---

**Algorithm 2** `TreeSwap.Alg`$(\mathcal{P}, \mathcal{L}, T, H, L)$, labeled variant (see Appendix C.1)

---

**Require:** Action set $\mathcal{P} \subset \mathbb{R}^d$, convex loss class $\mathcal{L} \subset (\mathcal{P} \to \mathbb{R})$, no-external regret algorithm `Alg`, time horizon $T$, parameters $H, L$ with $T \leq H^L$.

1:  For each sequence $h_1 \cdots h_{l-1} \in \bigcup_{l=1}^{L} [0 : H-1]^{l-1}$, initialize an instance of `Alg` with time horizon $H$, denoted $\mathtt{Alg}_{h_{1:l-1}}$.

2: **for** $1 \leq t \leq T$ **do**

3:      Write the base-$H$ representation of $t-1$ as $t-1 = (h_1 \cdots h_L)$, for $h_1, \ldots, h_L \in [0 : H-1]$.

4:      **for** $1 \leq l \leq L$ **do**

5:          Write $k := (h_1 \cdots h_{l-1}) \in [0 : H-1]^{l-1}$.

6:          **if** $h_{l+1} = \cdots = h_L = 0$ or $l = L$ **then**

7:              If $h_l > 0$, define $\ell_{k,h_l-1}^{(l)} := \frac{1}{H^{L-l}} \cdot \sum_{s \in \Gamma_{k,h_l-1}^{(l)}} \ell_s \in \mathcal{L}$.

8:              Define $p_{k,h_l}^{(l)} = \mathtt{Alg}_{k,h_l+1}(\ell_{k,0:h_l-1}^{(l)}) \in \mathcal{P}$.  ▷ *The $h_l$th action of $\mathtt{Alg}_k$ given the loss sequence $\ell_{k,1:h_l-1}^{(l)}$.*

9:          **end if**

10:      **end for**

11:      Output the uniform mixture $\mathbf{x}_t := \mathrm{Unif}(\{(p_{h_1}^{(1)}, h_1), \ldots, (p_{h_1 \cdots h_L}^{(L)}, h_{1:L})\}) \in \Delta(\bar{\mathcal{P}})$, and observe $\ell_t$.                    ▷ *Each action $p_k^{(l)}$ is labeled by the sequence $k$ (see Appendix C.1).*

12: **end for**

---

## C.2  Proof of the main theorem

First, we recall some definitions from Section 3. For all $l \in [0 : L]$, for all $k \in [H]^l$, let $\Gamma_k^{(l)}$ represent the interval of times $t$ with prefix $k$. That is, $t \in \Gamma_k^{(l)}$ iff $t_i = k_i$ for all $i \in [1 : l]$. These intervals form an $H$-ary depth-$L$ tree, where the children of $\Gamma_k^{(l)}$ are $\Gamma_{k0}^{(l+1)}, \Gamma_{k1}^{(l+1)}, \cdots, \Gamma_{k(H-1)}^{(l+1)}$. In the calibration setting where the learner receives outcomes $y_{1:T}$, let $\nu_k^{(l)} = \frac{1}{\left|\Gamma_k^{(l)}\right|} \sum_{t \in \Gamma_k^{(l)}} y_t$ (as defined on Line 6 of Algorithm 1). In the swap regret setting where the learner receives loss functions $\ell_{1:T}$, let $\ell_k^{(l)} = \frac{1}{\left|\Gamma_k^{(l)}\right|} \sum_{t \in \Gamma_k^{(l)}} \ell_t$ (as defined in Line 7 of Algorithm 2).

Finally, recall that for an online learning algorithm `Alg` with time horizon $H$, we define its action at time step $h \in [H]$ given losses $\ell_1, \ldots, \ell_H : \mathcal{P} \to \mathbb{R}$ by $\mathtt{Alg}_h(\ell_1, \ldots, \ell_H)$. If $\mathtt{Alg}_h$ only depends on the first $g$ losses, then we will write $\mathtt{Alg}_h(\ell_1, \ldots, \ell_g)$. In the proof of Theorem 3.1 we will consider two algorithms in particular; the first, Follow-The-Leader (FTL) is defined as follows: for

---

[10]Technically, the analysis of [DDFG24] does not analyse the labeled version, but the proof goes through as is – the only step where labeling changes any of the reasoning in the argument is in Eq. (8) of [DDFG24], where the upper bound as written in that equation holds even for the labeled version.

$\ell_1, \ldots, \ell_{h-1} : \mathcal{P} \to \mathbb{R}$, we have

$$\mathrm{FTL}_h(\ell_1, \ldots, \ell_{h-1}) = \operatorname*{argmin}_{p \in \mathcal{P}} \sum_{i=1}^{h-1} \ell_i(p).$$

The second algorithm we consider is the Be-The-Leader algorithm (BTL), which is defined as follows: for $\ell_1, \ldots, \ell_h : \mathcal{P} \to \mathbb{R}$, we have

$$\mathrm{BTL}_h(\ell_1, \ldots, \ell_h) = \operatorname*{argmin}_{p \in \mathcal{P}} \sum_{i=1}^{h} \ell_i(p).$$

Note that since $\mathrm{BTL}_h(\ell_{1:h})$ depends on the unobserved loss $\ell_h$ at time step $h$, it is unimplementable. Nevertheless, it will be useful in our analysis.

Next we prove Lemma 3.2, establishing the equivalence of `TreeCal` and `TreeSwap.FTL`. In fact, we establish the stronger claim, which immediately implies Lemma 3.2.

**Lemma C.2.** *Fix distributions $q_0, \ldots, q_h \in \Delta(\mathcal{P})$, and define $\ell_h(p) := \mathbb{E}_{y \sim q_h}[D_R(y|p)]$. Then for each $h > 0$, $\mathrm{FTL}_h(\ell_0, \ldots, \ell_{h-1}) = \frac{1}{h} \sum_{i=0}^{h-1} \mathbb{E}_{y \sim q_i}[y]$.*

*Proof.* The lemma is an immediate consequence of Lemma 2.1, noting that

$$\mathrm{FTL}_h(\ell_0, \ldots, \ell_{h-1}) = \operatorname*{argmin}_{p \in \mathcal{P}} \sum_{i=0}^{h-1} \ell_i(p) = \operatorname*{argmin}_{p \in \mathcal{P}} \mathbb{E}_{i \sim [0:h-1], y \sim q_i}[D_R(y_i|p)] = \frac{1}{h} \sum_{i=0}^{h-1} \mathbb{E}_{y \sim q_i}[y].$$

$$(7)$$

$\square$

*Proof of Lemma 3.2.* At time $t$, both `TreeCal` (Line 10 of Algorithm 1) and `TreeSwap.FTL` (Line 11 of Algorithm 2) select $\mathbf{x}_t = \mathrm{Unif}\left(\left\{p_{t_1}^{(1)}, p_{t_1 t_2}^{(2)}, \cdots, p_{t_1 t_2 \cdots t_L}^{(L)}\right\}\right)$. It remains to demonstrate that both algorithms assign actions $p_k^{(l)}$ to intervals $\Gamma_k^{(l)}$ identically. Fixing a choice of $l \in [L]$ and $k \in [0 : H-1]^{l-1}$, this is an immediate consequence of Lemma C.2 with $q_h = \mathrm{Unif}(\{y_t : t \in \Gamma_{k,h}^{(l)}\})$ and the fact that:

- In `TreeCal`, $p_{k,h}^{(l)} = \frac{1}{h} \sum_{i=0}^{h-1} \nu_{k,i}^{(l)}$ with $\nu_{k,i}^{(l)} = \mathbb{E}_{t \sim \mathrm{Unif}(\Gamma_{k,i}^{(l)})}[y_t]$;

- Whereas in `TreeSwap.FTL`, $p_{k,h}^{(l)} = \mathrm{FTL}_{h+1}(\ell_{k,0}^{(l)}, \ldots, \ell_{k,h-1}^{(l)})$ with $\ell_{k,i}^{(l)} = \mathbb{E}_{t \sim \mathrm{Unif}(\Gamma_{k,i}^{(l)})}[D_R(y_t|\cdot)]$.

$\square$

We are now ready to prove Theorem 3.1.

*Proof of Theorem 3.1.* Fix any convex set $\mathcal{P}$ and a norm $\|\cdot\|$, and let $R : \mathcal{P} \to \mathbb{R}$ be chosen to be 1-strongly convex which has range $\rho > 0$. Lemma 3.2 gives that the actions $\mathbf{x}_1, \ldots, \mathbf{x}_T \in \Delta(\mathcal{P})$ are identical to the actions played by `TreeSwap.FTL` with losses $\ell_t(p) = D_R(y_t|p)$ (Algorithm 2; we are ignoring the labels here). Thus, from here on, it suffices to bound the calibration error of the corresponding distributions $\mathbf{x}_1, \ldots, \mathbf{x}_T$ of `TreeSwap.FTL`. The actions $p_{k,h}^{(l)}$ (for $l \in [L], k \in [0 : H-1]^{l-1}, h \in [0 : H-1]$) of `TreeSwap.FTL` satisfy $p_{k,h}^{(l)} = \mathrm{FTL}_{h+1}(\ell_{k,0}^{(l)}, \ldots, \ell_{k,h-1}^{(l)})$.

Next, let $\tilde{p}_{k,h}^{(l)}$ denote the corresponding actions played by `TreeSwap.BTL`, i.e., $\tilde{p}_{k,h}^{(l)} = \mathrm{BTL}_{h+1}(\ell_{k,0}^{(l)}, \ldots, \ell_{k,h}^{(l)})$. We let $\mathbf{x}_t \in \Delta(\bar{\mathcal{P}})$ denote the (labeled) distribution chosen by `TreeSwap.FTL` (Line 11 of Algorithm 2), and let $\tilde{\mathbf{x}}_t \in \Delta(\bar{\mathcal{P}})$ denote the corresponding distribution for `TreeSwap.BTL`. To be concrete, if $t - 1 = (h_1 \cdots h_L)$, then

$$\mathbf{x}_t = \mathrm{Unif}(\{(p_{h_1}^{(1)}, h_1), \ldots, (p_{h_1 \cdots h_L}^{(L)}, h_{1:L})\}), \qquad \tilde{\mathbf{x}}_t = \mathrm{Unif}(\{(\tilde{p}_{h_1}^{(1)}, h_1), \ldots, (\tilde{p}_{h_1 \cdots h_L}^{(L)}, h_{1:L})\}), .$$

$$(8)$$

We state the below claim, whose proof is deferred to the end of the section. (We remark that the primary purpose of introducing labeling is so that it is possible to establish Claim C.3.)

**Claim C.3.** *It holds that*

$$\mathsf{Cal}_T^{\|\cdot\|^2}(\mathbf{x}_{1:T}, y_{1:T}) - 2\mathsf{Cal}_T^{\|\cdot\|^2}(\tilde{\mathbf{x}}_{1:T}, y_{1:T}) \leq \frac{2 \cdot \mathsf{diam}(\mathcal{P})^2}{H^2} \cdot T. \tag{9}$$

The fact that BTL enjoys non-positive external regret (e.g., [SS11, Lemma 2.1] gives that for an arbitrary sequence of loss functions $\ell_t : \mathcal{P} \to \mathbb{R}$, the external regret of $\mathtt{BTL}_H$ satisfies $\mathsf{ExtReg}_H(\mathtt{BTL}_H) \leq 0$. Thus, by Theorem C.1, the swap regret of (the labeled version of) $\mathtt{TreeSwap}_T$ applied with $\mathtt{Alg}_H = \mathtt{BTL}_H$ may be bounded as follows: for any sequence of losses $\ell_1, \dots, \ell_T : \mathcal{P} \to [0, \rho]$,

$$\mathsf{FullSwapReg}_T(\tilde{\mathbf{x}}_{1:T}, \ell_{1:T}) \leq T \cdot \frac{3\rho}{L}.$$

Using Lemma 2.2[11] and (9), we get that for an arbitrary sequence $y_1, \dots, y_T \in \mathcal{P}$,

$$\begin{aligned}
\mathsf{Cal}_T^{\|\cdot\|^2}(\mathbf{x}_{1:T}, y_{1:T}) &\leq 2 \cdot \mathsf{Cal}_T^{\|\cdot\|^2}(\tilde{\mathbf{x}}_{1:T}, y_{1:T}) + \frac{2 \cdot \mathsf{diam}(\mathcal{P})^2}{H^2} \cdot T \\
&\leq 2 \cdot \mathsf{Cal}_T^{D_R}(\tilde{\mathbf{x}}_{1:T}, y_{1:T}) + \frac{2 \cdot \mathsf{diam}(\mathcal{P})^2}{H^2} \cdot T \\
&= 2 \cdot \mathsf{FullSwapReg}_T(\tilde{\mathbf{x}}_{1:T}, D_R(y_{1:T}|\cdot)) + \frac{2 \cdot \mathsf{diam}(\mathcal{P})^2}{H^2} \cdot T \\
&\leq \frac{6\rho \cdot T}{L} + \frac{2 \cdot \mathsf{diam}(\mathcal{P})^2 \cdot T}{H^2}.
\end{aligned}$$

Given any desired accuracy $\epsilon > 0$, choosing $L = 12\rho/\epsilon$ and $H = \mathsf{diam}(\mathcal{P})/\sqrt{\epsilon}$ gives that we can guarantee $\mathsf{Cal}_T^{\|\cdot\|^2}(\mathbf{x}_{1:T}, y_{1:T}) \leq \epsilon \cdot T$ as long as $T \geq H^L = (\mathsf{diam}(\mathcal{P})/\sqrt{\epsilon})^{12\rho/\epsilon^2}$. $\qquad\square$

*Proof of Claim C.3.* For each $t \in [T]$, we can write $t - 1 = h_1 h_2 \cdots h_L$ with $h_i \in [0 : H - 1]$ for all $i \in [L]$, and $\mathbf{x}_t, \tilde{\mathbf{x}}_t$ are as given in (8). Let us write, for $(p, \sigma) \in \bar{\mathcal{P}}$,

$$\nu_{(p,\sigma)} := \frac{\sum_{t=1}^T \mathbf{x}_t((p, \sigma)) \cdot y_t}{\sum_{t=1}^T \mathbf{x}_t((p, \sigma))}, \qquad \tilde{\nu}_{(p,\sigma)} := \frac{\sum_{t=1}^T \tilde{\mathbf{x}}_t((p, \sigma)) \cdot y_t}{\sum_{t=1}^T \mathbf{x}_t((p, \sigma))}, \tag{10}$$

$$\nu_\sigma := \frac{\sum_{p \in \mathcal{P}} \sum_{t=1}^T \mathbf{x}_t((p, \sigma)) \cdot y_t}{\sum_{p \in \mathcal{P}} \sum_{t=1}^T \mathbf{x}_t((p, \sigma))}.$$

Since each $p_{h_1 \cdots h_l}^{(l)}$ and each $\tilde{p}_{h_1 \cdots h_l}^{(l)}$ is labeled by $h_{1:l}$ in $\mathbf{x}_t$ and $\tilde{\mathbf{x}}_t$, respectively, it holds that for each $\sigma$ of the form $\sigma = h_1 \cdots h_l$ (for some $l \in [L]$), there are unique $p, \tilde{p} \in \mathcal{P}$ so that $\nu_\sigma = \nu_{(p,\sigma)} = \nu_{(\tilde{p},\sigma)}$:

---

[11]Technically, we need a labeled version of Lemma 2.2, where the distribution $\mathbf{x}_t$ are over the labeled set $\Delta(\bar{\mathcal{P}})$; it is immediate to see that the proof of Lemma 2.2 extends to the labeled case.

in particular, we have $p = p^{(l)}_{h_1 \cdots h_l}, \tilde{p} = \tilde{p}^{(l)}_{h_1 \cdots h_l}$. We can therefore bound

$$
\mathsf{Cal}^{\|\cdot\|^2}_T(\mathbf{x}_{1:T}, y_{1:T}) - 2\mathsf{Cal}^{\|\cdot\|^2}_T(\tilde{\mathbf{x}}_{1:T}, y_{1:T})
$$

$$
= \sum_{l \in [L], h_{1:l} \in [0:H-1]^l} \left( \sum_{t=1}^{T} \mathbf{x}_t((p^{(l)}_{h_1 \cdots h_l}, h_1 \cdots h_l)) \right) \cdot \|\nu_{h_1 \cdots h_l} - p^{(l)}_{h_1 \cdots h_l}\|^2 \tag{11}
$$

$$
\qquad - 2 \left( \sum_{t=1}^{T} \tilde{\mathbf{x}}_t((\tilde{p}^{(l)}_{h_1 \cdots h_l}, h_1 \cdots h_l)) \right) \cdot \|\nu_{h_1 \cdots h_l} - \tilde{p}^{(l)}_{h_1 \cdots h_l}\|^2
$$

$$
= \sum_{l \in [L], h_{1:l} \in [0:H-1]^l} \frac{H^{L-l}}{L} \cdot \left( \left\| \nu_{h_1 \cdots h_l} - p^{(l)}_{h_1 \cdots h_l} \right\|^2 - 2 \left\| \nu_{h_1 \cdots h_l} - \tilde{p}^{(l)}_{h_1 \cdots h_l} \right\|^2 \right)
$$

$$
\leq 2 \sum_{l \in [L], h_{1:l} \in [0:H-1]^l} \frac{H^{L-l}}{L} \cdot \left\| p^{(l)}_{h_1 \cdots h_l} - \tilde{p}^{(l)}_{h_1 \cdots h_l} \right\|^2
$$

$$
\leq \frac{2}{L} \sum_{l=1}^{L} \sum_{h_{1:l-1} \in [0:H-1]^{l-1}} \mathsf{diam}(\mathcal{P})^2 \cdot H^{L-l}
$$

$$
\leq \frac{2}{L} \sum_{l=1}^{L} \mathsf{diam}(\mathcal{P})^2 \cdot H^{L-1}
$$

$$
= \frac{2T\mathsf{diam}(\mathcal{P})^2}{H},
$$

where the second-to-last inequality uses that $\sum_{h_l=0}^{H-1} \left\| p^{(l)}_{h_1 \cdots h_l} - \tilde{p}^{(l)}_{h_1 \cdots h_l} \right\|^2 \leq \mathsf{diam}(\mathcal{P})^2$ for all choices of $h_1 \cdots h_{l-1}$ (a consequence of Lemma C.4 and Lemma C.2). $\qquad \square$

**Lemma C.4.** *Fix any convex set $\mathcal{P} \subset \mathbb{R}^d$ and a convex function $R : \mathcal{P} \to \mathbb{R}$. Fix a sequence $y_1, \ldots, y_H \in \mathcal{P}$, and set*

$$
p_h = \frac{1}{h-1} \sum_{i=1}^{h-1} y_i \ \forall h \in [H], h > 1, \qquad \tilde{p}_h = \frac{1}{h} \sum_{i=1}^{h} y_i \ \forall h \in [H],
$$

*as well as $p_1 \in \mathcal{P}$ arbitrarily. Then*

$$
\sum_{h=1}^{H} \|p_h - \tilde{p}_h\|^2 \leq 2 \cdot \mathsf{diam}(\mathcal{P})^2.
$$

*Proof.* Note that

$$
\tilde{p}_h - p_h = \frac{y_h}{h} - \frac{1}{h(h-1)} \sum_{i=1}^{h-1} y_i,
$$

which implies that $\|\tilde{p}_h - p_h\|^2 \leq \frac{\pi^2}{6} \cdot \mathsf{diam}(\mathcal{P})^2 < 2\mathsf{diam}(\mathcal{P})^2$. $\qquad \square$

Applying Cauchy-Schwarz, we get the following corollary,

**Corollary C.5.** *Let $\mathcal{P} \subset \mathbb{R}^d$ be a bounded convex set and $\|\cdot\|$ be an arbitrary norm. Then, TreeCal (Algorithm 1) guarantees that for an arbitrary sequence of outcomes $y_1, \ldots, y_T \in \mathcal{P}$, the $\|\cdot\|$ calibration error of its predictions $\mathbf{x}_1, \ldots, \mathbf{x}_T \in \Delta(\mathcal{P})$ is bounded $\mathsf{Cal}^{\|\cdot\|}_T(\mathbf{x}_{1:T}, y_{1:T}) \leq \epsilon T$ for $T \geq (\mathsf{diam}_{\|\cdot\|}(\mathcal{P})/\epsilon)^{O(\mathsf{Rate}(\mathcal{P}, \|\cdot\|)/\epsilon^2)}$*

*Proof.* Using the fact that $\sum_{p \in \mathcal{P}} \sum_{t=1}^{T} \mathbf{x}_t(p) = 1$ together with Jensen's inequality, we have

$$\frac{1}{T}\mathsf{Cal}_T^{\|\cdot\|}(\mathbf{x}_{1:T}, y_{1:T}) = \frac{1}{T} \sum_{p \in \mathcal{P}} \left( \sum_{t=1}^{T} \mathbf{x}_t(p) \right) \cdot \|\nu_p - p\|$$

$$\leq \sqrt{\frac{1}{T} \sum_{p \in \mathcal{P}} \left( \sum_{t=1}^{T} \mathbf{x}_t(p) \right) \cdot \|\nu_p - p\|^2}$$

$$= \sqrt{\frac{1}{T}\mathsf{Cal}_T^{\|\cdot\|^2}(\mathbf{x}_{1:T}, y_{1:T})} \leq \epsilon$$

for $T \geq (\mathsf{diam}(\mathcal{P})/\epsilon)^{O(\mathsf{Rate}(\mathcal{P},\|\cdot\|)/\epsilon^2)}$ by Theorem 3.1. Thus, $\mathsf{Cal}_T^{\|\cdot\|}(\mathbf{x}_{1:T}, y_{1:T}) \leq \epsilon T$ for $T \geq (\mathsf{diam}(\mathcal{P})/\epsilon)^{O(\mathsf{Rate}(\mathcal{P},\|\cdot\|)/\epsilon^2)}$, incurring an additional factor of 2 in the exponent constant, as desired. $\square$

Finally, for the setting of centrally symmetric $\mathcal{P}$, we can apply Lemma 2.4 to directly relate this regret bound to the optimal possible rate of an online linear optimization problem.

**Corollary C.6.** *Let $\mathcal{P} \subset \mathbb{R}^d$ be a bounded centrally symmetric convex set and $\|\cdot\|$ be an arbitrary norm. Then, $\mathtt{TreeCal}$ (Algorithm 1) guarantees that for an arbitrary sequence of outcomes $y_1, \ldots, y_T \in \mathcal{P}$, the $\|\cdot\|$ calibration error of its predictions $\mathbf{x}_1, \ldots, \mathbf{x}_T \in \Delta(\mathcal{P})$ is bounded $\mathsf{Cal}_T^{\|\cdot\|}(\mathbf{x}_{1:T}, y_{1:T}) \leq \epsilon T$ for $T \geq (\mathsf{diam}_{\|\cdot\|}(\mathcal{P})/\epsilon)^{O(\mathsf{Rate}_{\mathsf{OLO}}(\mathcal{P},\|\cdot\|)/\epsilon^2)}$*

*Proof.* Follows immediately by applying Lemma 2.4 to Corollary C.5. $\square$

# D   Proofs for Section 4

In this section, we prove lower bounds on high-dimensional calibration that tell us that in order to achieve calibration error at most $\epsilon \cdot T$, we need to take $T \gtrsim \exp(\mathsf{poly}(1/\epsilon))$. First, in Appendix D.1, we prove a lower bound for $\ell_1$ calibration over the $d$-dimensional simplex, and then, in Appendix D.2, we prove a lower bound for $\ell_2$ calibration over the unit $d$-dimensional Euclidean ball.

## D.1   Lower bound on $\ell_1$ calibration

First, we prove Theorem 4.3 which gives a lower bound on $\ell_1$ calibration over the simplex $\mathcal{P} = \Delta^d$.

*Proof of Lemma 4.2.* Fix an algorithm $\mathtt{Alg}$ which ensures that $\mathsf{Cal}_T^D(\mathbf{x}_{1:T}, y_{1:T}) \leq \epsilon \cdot T$ as in the statement of the lemma. We construct the following algorithm $\mathtt{Alg}'$: it simulates $\mathtt{Alg}$, but whenever $\mathtt{Alg}$ outputs the distribution $\mathbf{x}_t \in \Delta(\mathcal{P})$, $\mathtt{Alg}'$ chooses instead $\mathbf{x}_t' \in \Delta(\mathcal{P}')$, defined by

$$\mathbf{x}_t'(p') := \sum_{\substack{p \in \mathcal{P}: \\ p' = \mathrm{argmin}_{q \in \mathcal{P}'} \langle q, p \rangle}} \mathbf{x}_t(p).$$

To simplify notation, we define $\mathsf{BR}(p) := \operatorname{argmin}_{q \in \mathcal{P}'} \langle q, p \rangle$. It follows that, for any oblivious adversary choosing a (random) sequence $y_1, \dots, y_T \in \mathcal{P}$,

$$\mathsf{FullSwapReg}_T(\mathbf{x}'_{1:T}, \ell(\cdot, y_{1:T}))$$

$$= \sup_{\pi:\mathcal{P}'\to\mathcal{P}'} \sum_{p'\in\mathcal{P}} \sum_{t\in[T]} \mathbf{x}'_t(p') \cdot (\langle y_t, p' - \pi(p')\rangle)$$

$$= \sup_{\pi:\mathcal{P}'\to\mathcal{P}'} \sum_{p\in\mathcal{P}} \sum_{t\in[T]} \mathbf{x}_t(p) \cdot (\langle y_t, \mathsf{BR}(p) - \pi(\mathsf{BR}(p))\rangle)$$

$$= \sup_{\pi:\mathcal{P}'\to\mathcal{P}'} \sum_{p\in\mathcal{P}} \left( \sum_{t\in[T]} \mathbf{x}_t(p) \right) \cdot (\langle \nu_p, \mathsf{BR}(p) - \pi(\mathsf{BR}(p))\rangle)$$

$$= \sup_{\pi:\mathcal{P}'\to\mathcal{P}'} \sum_{p\in\mathcal{P}} \left( \sum_{t\in[T]} \mathbf{x}_t(p) \right) \cdot (\langle \nu_p - p, \mathsf{BR}(p) - \pi(\mathsf{BR}(p))\rangle + \langle p, \mathsf{BR}(p) - \pi(\mathsf{BR}(p))\rangle)$$

$$\leq \sup_{\pi:\mathcal{P}'\to\mathcal{P}'} \sum_{p\in\mathcal{P}} \left( \sum_{t\in[T]} \mathbf{x}_t(p) \right) \cdot (\|\nu_p - p\| \cdot \|\mathsf{BR}(p) - \pi(\mathsf{BR}(p))\|_\star)$$

$$\leq \mathsf{diam}_{\|\cdot\|_\star}(\mathcal{P}') \cdot \mathsf{Cal}_T^D(\mathbf{x}_{1:T}, y_{1:T}),$$

where in the final inequality we have used the fact that $\|\mathsf{BR}(p) - \pi(\mathsf{BR}(p))\|_\star \leq \mathsf{diam}_{\|\cdot\|_\star}(\mathcal{P}')$. $\qquad\square$

For $p > 0$, $d \in \mathbb{N}$, write $\mathcal{B}_p^d := \{x \in \mathbb{R}^d \mid \|x\|_p \leq 1\}$ to denote the unit $\ell_p$ norm ball.

To map the lower bound Theorem 4.1 from the $\|\cdot\|_1$-norm unit ball $\mathcal{B}_1^d$ to the simplex and arrive at the desired contradiction using the above lemma, we use the following.

**Lemma D.1.** *Fix $d \in \mathbb{N}$, and write $D(x, y) := \|x - y\|_1$ for $x, y \in \mathbb{R}^d$. Suppose that there is an algorithm* $\mathtt{Alg}$ *for calibration over the domain $\mathcal{P} = \Delta^{2d+1}$ which produces $\mathbf{x}_{1:T}$ given the choices of an adversary $y_{1:T}$ achieving calibration error $\mathsf{Cal}_T^D(\mathbf{x}_{1:T}, y_{1:T}) \leq R(T)$, for $T \in \mathbb{N}$. Then there is an algorithm* $\mathtt{Alg}'$ *for calibration over the domain $\mathcal{B}_1^d$ which produces $\mathbf{x}'_{1:T}$ given $y'_{1:T}$ achieving calibration error $\mathsf{Cal}_T^D(\mathbf{x}'_{1:T}, y'_{1:T}) \leq R(T)$.*

*Proof of Lemma D.1.* We define a mapping $\phi : \mathcal{B}_1^d \to \Delta^{2d+1}$ as follows: for $y \in \mathcal{B}_1^d \subset \mathbb{R}^d$, we define

$$\phi(y)_i = \begin{cases} [y_j]_+ & : i = 2j-1, \ j \in [d] \\ [y_j]_- & : i = 2j, \ j \in [d] \\ 1 - \|y\| & : i = 2d+1. \end{cases}$$

It is straightforward to see that $\phi$ has a left inverse $\psi$, defined as follows: for $z \in \Delta^{2d+1}$,

$$\psi(z)_i = z_{2i-1} - z_{2i}, \quad i \in [d],$$

so that $\psi \circ \phi(y) = y$ for all $y \in \mathbb{R}^d$.

We define the algorithm $\mathtt{Alg}'$ as follows: given $y'_t \in \mathcal{B}_1^d \subset \mathbb{R}^d$, it defines $y_t \in \Delta^{2d+1}$ by $y_t = \phi(y'_t)$. $\mathtt{Alg}'$ then feeds $y_t$ into $\mathtt{Alg}$, and if we denote the distribution output by $\mathtt{Alg}$ at time step $t$ by $\mathbf{x}_t$, $\mathtt{Alg}'$ then plays the push-forward measure $\mathbf{x}'_t := \psi \circ \mathbf{x}_t \in \Delta(\mathcal{B}_1^d)$.

Our bound on the calibration error of $\mathtt{Alg}$ gives

$$\mathsf{Cal}_T^D(\mathbf{x}_{1:T}, y_{1:T}) = \sum_{p \in \Delta^{2d+1}} \left( \sum_{t=1}^T \mathbf{x}_t(p) \right) \cdot \|\nu_p - p\|_1 \leq R(T),$$

where $\nu_p = \frac{\sum_{t=1}^T \mathbf{x}_t(p) \cdot y_t}{\sum_{t=1}^T \mathbf{x}_t(p)} \in \Delta^{2d+1}$. For $p' \in \mathcal{B}_1^d$, let us denote $\nu'_{p'} := \frac{\sum_{t=1}^T \mathbf{x}'_t(p') \cdot y'_t}{\sum_{t=1}^T \mathbf{x}'_t(p')} = \psi\left( \frac{\sum_{t=1}^T \mathbf{x}'_t(p') \cdot y_t}{\sum_{t=1}^T \mathbf{x}'_t(p')} \right)$, using linearity of $\psi$.

We may now bound the calibration error of $\texttt{Alg}'$ by

$$\begin{aligned}
\mathsf{Cal}_T^D(\mathbf{x}'_{1:T}, y'_{1:T}) &= \sum_{p' \in \mathcal{B}_1^d} \left( \sum_{t=1}^T \mathbf{x}'_t(p') \right) \cdot \|\nu'_{p'} - p'\|_1 \\
&\leq \sum_{p \in \Delta^{2d+1}} \left( \sum_{t=1}^T \mathbf{x}_t(p) \right) \cdot \|\psi(\nu_p) - \psi(p)\|_1 \\
&\leq \mathsf{Cal}_T^D(\mathbf{x}_{1:T}, y_{1:T}).
\end{aligned}$$

$\square$

*Proof of Theorem 4.3.* Suppose to the contrary that there was an algorithm $\texttt{Alg}$ which bounded calibration error by $\epsilon T$ for $T \leq \exp(c \cdot \min\{d^{1/14}, \epsilon^{-1/6}\})$. Then by Lemma D.1, for $d' = \lfloor (d-1)/2 \rfloor$ there is an algorithm $\texttt{Alg}'$ for calibration on the domain $\mathcal{B}_1^{d'} \subset \mathbb{R}^{d'}$ produces $\mathbf{x}'_{1:T}$ given $y'_{1:T}$ satisfying $\mathsf{Cal}_T^D(\mathbf{x}'_{1:T}, y'_{1:T}) \leq \epsilon \cdot T$ for any $T \leq \exp(c \cdot \min\{d^{1/14}, \epsilon^{-1/6}\})$.

We now apply Lemma 4.2 for $\mathcal{P} = \mathcal{B}_1^{d'} \subset \mathbb{R}^{d'}$, the norm given by the $\ell_1$ norm $\| \cdot \|_1$, and $\mathcal{P}' := [-1, 1]^{d'}$. Note that $\mathsf{diam}_{\|\cdot\|_\infty}(\mathcal{P}') = 1$. Then Lemma 4.2 ensures that there is an algorithm $\texttt{Alg}''$ which chooses $\mathbf{x}''_1, \ldots, \mathbf{x}''_T \in \Delta(\mathcal{P}')$ which ensures that for every oblivious adversary choosing $y''_1, \ldots, y''_T \in \mathcal{B}_1^{d'}$, we have $\mathsf{FullSwapReg}_T(\mathbf{x}''_{1:T}, \ell(\cdot, y''_{1:T})) \leq \epsilon \cdot T$.

But if $T \leq \exp(c_{4.1} \cdot \min\{(d')^{1/14}, \epsilon^{-1/6}\})$, we have a contradiction to Theorem 4.1, thus completing the proof of the theorem. $\square$

## D.2 Lower bound for $\ell_2$ calibration

Next, we prove a lower bound for $\ell_2$ calibration.

**Theorem D.2.** *There is a sufficiently small constant $c > 0$ so that the following holds. Write $D(p, p') = \|p - p'\|_2$ and fix any $\epsilon > 0$, $d \in \mathbb{N}$. Then for any $T \leq \exp(c \cdot \min\{d^{1/14}, \epsilon^{-1/7}\})$, there is an oblivious adversary producing a sequence $y_1, \ldots, y_T \in \mathcal{B}_2^d$ so that for any learning algorithm producing $\mathbf{x}_1, \ldots, \mathbf{x}_T \in \Delta(\mathcal{B}_2^d)$, $\mathsf{Cal}_T^D(\mathbf{x}_{1:T}, y_{1:T}) \geq \epsilon \cdot T$.*

*Proof.* Fix $\epsilon > 0, d \in \mathbb{N}$, and write $\tilde{\epsilon} = \epsilon^{6/7}$. We may assume without loss of generality that $d \leq \tilde{\epsilon}^{-14/6}$, so that $\min\{d^{1/14}, \tilde{\epsilon}^{-1/6}\} = \min\{d^{1/14}, \epsilon^{-1/7}\} = d^{1/14}$: if this were not the case, we simply use the adversary resulting from $\tilde{\epsilon}^{-14/6}$ dimensions and project the forecaster's predictions down into this lower-dimensional subspace, which can only decrease calibration error. Now suppose to the contrary that there was an algorithm $\texttt{Alg}$ which bounded calibration error by $\epsilon T$ for $T \leq \exp(c \cdot \min\{d^{1/14}, \epsilon^{-1/7}\}) = \exp(c \cdot d^{1/14})$. Then by Lemma 4.2 with $\mathcal{P} = \mathcal{B}_2^d$ and norm $\|\cdot\| = \|\cdot\|_2$, for any subset $\mathcal{P}' \subset \mathcal{B}_2^d$ we get that there is an algorithm which chooses $\mathbf{x}'_1, \ldots, \mathbf{x}'_T \in \Delta(\mathcal{P}')$ and which ensures that for every oblivious adversary choosing $y_1, \ldots, y_T \in \mathcal{B}_2^d$, we have

$$\mathsf{FullSwapReg}_T(\mathbf{x}'_{1:T}, (\langle \cdot, y_1 \rangle, \ldots, \langle \cdot, y_T \rangle)) \leq \epsilon \cdot T. \tag{12}$$

On the other hand, the oblivious adversary of Theorem 4.1 guarantees a subset $\mathcal{X} \subset [-1, 1]^d \subset$ and an oblivious adversary producing a sequence $v_1, \ldots, v_T \in \mathbb{R}^d$ with $\|v_t\|_\infty \leq d^{-13/14}$ for all $t \in [T]$, so that

$$\mathsf{FullSwapReg}_T(\mathbf{x}_{1:T}, (\langle \cdot, v_1 \rangle, \ldots, \langle \cdot, v_T \rangle)) \geq \tilde{\epsilon} \cdot T \tag{13}$$

as long as $T \leq \exp(c_{4.1} \cdot d^{1/14})$. We have $\|v_t\|_2 \leq d^{1/2 - 13/14} = d^{-3/7}$ for all $t$, and scaling $\mathcal{X}$ down by a factor of $1/\sqrt{d}$ (i.e., letting $\mathcal{P}' = \mathcal{X}/\sqrt{d}$) and all vectors $v_t$ up by a factor of $d^{3/7}$ (i.e., letting $v'_t = \sqrt{d} \cdot v_t$ ensures that any algorithm producing $\mathbf{x}'_1, \ldots, \mathbf{x}'_T \in \mathcal{P}'$ must still have full swap regret

$$\mathsf{FullSwapReg}_T(\mathbf{x}'_{1:T}, (\langle \cdot, v'_1 \rangle, \ldots, \langle \cdot, v'_T \rangle)) > \tilde{\epsilon} \cdot T \cdot d^{-1/14} \geq \tilde{\epsilon}^{7/6} \cdot T = \epsilon \cdot T,$$

but now ensures that $\mathcal{P}' \subset \mathcal{B}_2^d$ and that $v'_t \in \mathcal{B}_2^d$ for all $t$. By taking $c = c_{4.1}$, this contradicts (12). $\square$

# E  Pure calibration and pure full swap regret

## E.1  Pure calibration

In certain settings of calibration, the learner is required to randomly select a pure forecast $p_t \in \mathcal{P}$ rather than a distribution $\mathbf{x}_t \in \Delta(\mathcal{P})$. In these settings, the above definition of calibration is instead referred to as "pseudo-calibration". Here, we stick to calling the above calibration, as we believe it to be the more natural definition, and instead refer to this alternative setting as "pure-calibration". The learning task changes as follows.

At each time step $t \in [T]$:

- The learning algorithm chooses a distribution $\mathbf{x}_t \in \Delta(\mathcal{P})$.
- The adversary observes $\mathbf{x}_t$ and chooses an *outcome* $y_t \in \mathcal{P}$.
- The learner samples $p_t \sim \mathbf{x}_t$.

We adjust the definitions of the "pure average outcome" and "pure-calibration" accordingly:

$$
\dot{\nu}_p := \frac{\sum_{t=1}^T \mathbb{1}[p_t = p] \cdot y_t}{\sum_{t=1}^T \mathbb{1}[p_t = p]}, \qquad \mathsf{PureCal}_T^D(p_{1:T}, y_{1:T}) := \sum_{p \in \mathcal{P}} \left( \sum_{t=1}^T \mathbb{1}[p_t = p] \right) \cdot D(\dot{\nu}_p, p)
$$

---

**Algorithm 3** $\mathtt{SampleTreeCal}(\mathcal{P}, T, H, L, S)$

---

**Require:** Action set $\mathcal{P} \subset \mathbb{R}^d$, time horizon $T$, repetition parameter $S$ parameters $H, L$ with $T/S \leq H^L$.

1: Instantiate an instance $\mathtt{TreeCal}(\mathcal{P}, T/S, H, L)$.
2: **for** $1 \leq i \leq T/S$ **do**
3:    Let $\mathbf{x}_i \in \Delta(\mathcal{P})$ denote the prediction of $\mathtt{TreeCal}$ at step $i$.
4:    **for** $1 \leq j \leq S$ **do**
5:        Sample $p_{S(i-1)+j} \sim \mathbf{x}_i$, and observe outcome $y_{S(i-1)+j}$.
6:    **end for**
7:    Feed the outcome $\bar{y}_i := \frac{1}{S} \sum_{j=1}^S y_{S(i-1)+j}$ to $\mathtt{TreeCal}$.
8: **end for**

---

To obtain a bound on the (expected) pure calibration error, we use a slight modification of $\mathtt{TreeCal}$, namely $\mathtt{SampleTreeCal}$ (Algorithm 3). It functions identically to $\mathtt{TreeCal}$ except that for each time step $t$ of $\mathtt{TreeCal}$, it samples $S$ actions from $\mathbf{x}_t$ on each of $S$ contiguous time steps. (Hence, $\mathtt{TreeCal}$ is used with time horizon $T/S$.) At a high level, we will use an appropriate concentration inequality to show that the calibration upper bound of Theorem 3.1 implies a *pure calibration* upper bound for $\mathtt{SampleTreeCal}$.

**Theorem E.1** (Pure calibration error). *Let $\mathcal{P} \subset \mathbb{R}^d$ be a bounded convex set and $\|\cdot\|$ be an arbitrary norm with unit dual ball $\mathcal{L} := \{f \in \mathbb{R}^d \mid \|f\|_\star \leq 1\}$. Then, $\mathtt{SampleTreeCal}$ (Algorithm 3, with an appropriate choice of parameters $H, L, S$) guarantees that for an arbitrary sequence of outcomes $y_1, \ldots, y_T \in \mathcal{P}$, the $\|\cdot\|^2$ calibration error of its predictions $\mathbf{x}_1, \ldots, \mathbf{x}_T \in \Delta(\mathcal{P})$ is bounded as follows:*

$$
\mathbb{E}[\mathsf{PureCal}_T^{\|\cdot\|}(p_{1:T}, y_{1:T})] \leq \epsilon T, \quad \text{for} \quad T \geq \mathsf{Rate}(\mathcal{L}, \|\cdot\|_\star) \cdot (\mathsf{diam}_{\|\cdot\|}(\mathcal{P})/\epsilon)^{O(\mathsf{Rate}(\mathcal{P}, \|\cdot\|)/\epsilon^2)}.
$$

*Proof.* The proof uses Theorem 3.1 together with an appropriate concentration inequality, and closely follows that of [Pen25, Lemma 3.4].

Fix any $1 \leq i \leq T/S$ and $1 \leq j \leq S$, and let $\mathscr{F}_{S(i-1)+j}$ denote the $\sigma$-algebra generated by $y_1, \ldots, y_{S(i-1)+j+1}$ and $p_1, \ldots, p_{S(i-1)+j}$; since $\mathtt{TreeCal}$ is deterministic, it follows that $\mathbf{x}_1, \ldots, \mathbf{x}_i \in \Delta(\mathcal{P})$ are $\mathscr{F}_i$-measurable. For any $1 \leq j \leq S$, we have that, for any $p \in \mathrm{supp}(\mathbf{x}_i)$,

$$
\mathbb{E}\left[(p - y_{S(i-1)+j}) \cdot \mathbb{1}[p_{S(i-1)+j} = p] \mid \mathscr{F}_{S(i-1)+j-1}\right] = (p - y_{S(i-1)+j}) \cdot \mathbf{x}_i(p).
$$

Fixing any $i \in [T/S]$, By Lemma E.2 applied to the sequence $M_{S(i-1)+j} = (p - y_{S(i-1)+j}) \cdot \mathbb{1}[p_{S(i-1)+j} = p]$, for $1 \leq j \leq S$ (and the filtration $\mathscr{F}_{S(i-1)+j}$), we see that

$$\mathbb{E}\left[\left\|\sum_{j=1}^{S}(p - y_{S(i-1)+j}) \cdot \mathbb{1}[p_{S(i-1)+j} = p] - \sum_{j=1}^{S}(p - y_{S(i-1)+j}) \cdot \mathbf{x}_i(p)\right\|\right] \leq \mathsf{diam}_{\|\cdot\|}(\mathcal{P}) \cdot \sqrt{8S \cdot \mathsf{Rate}(\mathcal{L}, \|\cdot\|_\star)}.$$

It follows by summing over the $L$ values of $p \in \mathrm{supp}(\mathbf{x}_i)$ that

$$\mathbb{E}\left[\sum_{p \in \mathcal{P}}\left\|\sum_{j=1}^{S}(p - y_{S(i-1)+j}) \cdot \mathbb{1}[p_{S(i-1)+j} = p] - \sum_{j=1}^{S}(p - y_{S(i-1)+j}) \cdot \mathbf{x}_i(p)\right\|\right]$$

$$\leq L \cdot \mathsf{diam}_{\|\cdot\|}(\mathcal{P}) \cdot \sqrt{8S \cdot \mathsf{Rate}(\mathcal{L}, \|\cdot\|_\star)} \leq \epsilon \cdot S, \tag{14}$$

as long as $S \geq \frac{8 \cdot \mathsf{Rate}(\mathcal{L}, \|\cdot\|_\star) \cdot \mathsf{diam}_{\|\cdot\|}(\mathcal{P})^2 \cdot L^2}{\epsilon^2}$.

The guarantee of Corollary C.5 gives that, as long as $T/S \geq (\mathsf{diam}_{\|\cdot\|}(\mathcal{P})/\epsilon)^{O(\mathsf{Rate}(\mathcal{P}, \|\cdot\|)/\epsilon^2)}$, then

$$\mathsf{Cal}_T^{\|\cdot\|}(\mathbf{x}_{1:T/S}, \bar{y}_{1:T/S}) = \sum_{p \in \mathcal{P}}\left\|\sum_{i=1}^{T/S}\mathbf{x}_i(p) \cdot (p - \bar{y}_i)\right\|$$

$$= \sum_{p \in \mathcal{P}}\left\|\sum_{i=1}^{T/S}\frac{1}{S}\sum_{j=1}^{S}\mathbf{x}_i(p) \cdot (p - y_{S(i-1)+j})\right\| \leq \frac{\epsilon T}{S}. \tag{15}$$

By combining Equations (14) and (15), it follows that for an arbitrary adaptive adversary who chooses a sequence $y_1, \ldots, y_T \in \mathcal{P}$,

$$\mathbb{E}\left[\mathsf{PureCal}_T^{\|\cdot\|}(p_{1:T}, y_{1:T})\right]$$

$$= \mathbb{E}\left[\sum_{p \in \mathcal{P}}\left\|\sum_{t=1}^{T}(p - y_t) \cdot \mathbb{1}[p_t = p]\right\|\right]$$

$$\leq \mathbb{E}\left[\sum_{p \in \mathcal{P}}\left\|\sum_{i=1}^{T/S}\sum_{j=1}^{S}(p - y_{S(i-1)+j}) \cdot \mathbf{x}_i(p)\right\| + \sum_{i=1}^{T/S}\left\|\sum_{j=1}^{S}\left((p - y_{S(i-1)+j}) \cdot (\mathbb{1}[p_{S(i-1)+j} = p] - \mathbf{x}_i(p))\right)\right\|\right]$$

$$\leq 2\epsilon T.$$

The result follows by rescaling $\epsilon$ and our choice of $L = O(\mathsf{Rate}(\mathcal{P}, \|\cdot\|)/\epsilon^2)$. $\qquad \square$

As example applications of Theorem E.1:

- When $\|\cdot\|$ is the $\ell_1$ norm and $\mathcal{P}$ is the simplex, we have $\mathsf{diam}_{\|\cdot\|}(\mathcal{P}) = 1$, $\mathcal{L} = \{f \in \mathbb{R}^d \mid \|f\|_\infty \leq 1\}$ satisfies $\mathsf{Rate}(\mathcal{L}, \|\cdot\|_\star) \leq d$ (as we can take the function $R(x) = \|x\|_2^2$), which gives that for $T \geq d^{O(1/\epsilon^2)}$, we can have $\mathbb{E}[\mathsf{PureCal}_T^{\|\cdot\|_1}] \leq \epsilon T$. This result recovers the main upper bound of [Pen25] (Theorem 1.1 therein).
- When $\|\cdot\|$ is the $\ell_2$ norm and $\mathcal{P}$ is the unit $\ell_2$ ball, we have $\mathsf{diam}_{\|\cdot\|}(\mathcal{P}) = 1$, $\mathcal{L} = \{f \in \mathbb{R}^d \mid \|f\|_2 \leq 1\}$ satisfies $\mathsf{Rate}(\mathcal{L}, \|\cdot\|_\star) \leq 1$ (as we can take the function $R(x) = \|x\|_2^2$), which gives that for $T \geq \exp(O(1/\epsilon^2))$, we can have $\mathbb{E}[\mathsf{PureCal}_T^{\|\cdot\|_1}] \leq \epsilon T$.

### E.2 Sequential law of large numbers

Fix a convex set $\mathcal{P} \subset \mathbb{R}^d$ and a norm $\|\cdot\|$ on $\mathbb{R}^d$. We define

$$\mathcal{R}_n(\mathcal{P}, \|\cdot\|) := \sup_{\mathbf{p}} \mathbb{E}_\epsilon\left[\left\|\frac{1}{n}\sum_{i=1}^{n}\epsilon_i \mathbf{p}_i(\epsilon)\right\|\right],$$

where the supremum is over all sequences of mappings $\mathbf{p}_1, \ldots, \mathbf{p}_n$, where $\mathbf{p}_i : \{-1, 1\}^{i-1} \to \mathcal{P}$, and the expectation is over an i.i.d. sequence of Rademacher random variables $\epsilon = (\epsilon_1, \ldots, \epsilon_n)$, $\epsilon_i \sim \mathrm{Unif}(\{\pm 1\})$. The below lemma (essentially contained in [RST15]) establishes a martingale law of large numbers for $\mathcal{P}$-valued martingales, in terms of geometric properties of $\mathcal{P}$ and $\|\cdot\|$.

**Lemma E.2** ([RST15]). *Consider a convex set $\mathcal{P} \subset \mathbb{R}^d$ a norm $\|\cdot\|$ on $\mathbb{R}^d$, and let $M_1, \ldots, M_n$ denote a sequence of random variables adapted to a filtration $(\mathscr{F}_i)_{i \in [n]}$. Let $\mathcal{L} = \{f \mid \|f\|_\star \leq 1\}$ be the unit ball of the dual norm $\|\cdot\|$. Then*

$$\mathbb{E}\left[\left\|\sum_{i=1}^n M_i - \mathbb{E}[M_i \mid \mathscr{F}_{i-1}]\right\|\right] \leq \mathsf{diam}_{\|\cdot\|}(\mathcal{P}) \cdot \sqrt{8n \cdot \mathsf{Rate}(\mathcal{L}, \|\cdot\|_\star)}.$$

*Proof.* By applying an appropriate translation to $\mathcal{P}$, we can assume that $\mathcal{P}$ contains the origin. We apply Theorem 2 of [RST15] with the domain $\mathcal{Z}$ equal to $\mathcal{P}$ and the function class $\mathcal{F}$ equal to the class of mappings $\{z \mapsto \langle z, f \rangle : \|f\|_\star \leq 1\}$ indexed by unit-dual norm linear functions on $\mathcal{Z}$. The theorem implies that

$$\mathbb{E}\left[\frac{1}{n}\left\|\sum_{i=1}^n M_i - \mathbb{E}[M_i \mid \mathscr{F}_{i-1}]\right\|\right] \leq 2 \cdot \sup_{\mathbf{p}} \mathbb{E}_\epsilon\left[\sup_{\|f\|_\star \leq 1} \frac{1}{n}\left\langle \sum_{i=1}^n \epsilon_i \mathbf{p}_i(\epsilon), f \right\rangle\right]$$
$$= 2 \cdot \mathcal{R}_n(\mathcal{P}, \|\cdot\|).$$

Write $\mathcal{L} = \{f \in \mathbb{R}^d : \|f\|_\star \leq 1\}$ denote the unit ball for the dual norm $\|\cdot\|_\star$. Proposition 16 of [RST15] gives that, if there is a function $R : \mathcal{L} \to \mathbb{R}$ which is 1-strongly convex with respect to $\|\cdot\|_\star$ and which has range $\rho$, then $\mathcal{R}_n(\mathcal{P}, \|\cdot\|) \leq \sqrt{\frac{2\rho}{n}} \cdot \mathsf{diam}_{\|\cdot\|}(\mathcal{P})$. In particular, $\mathcal{R}_n(\mathcal{P}, \|\cdot\|) \leq \mathsf{diam}_{\|\cdot\|}(\mathcal{P}) \cdot \sqrt{\frac{2\mathsf{Rate}(\mathcal{L}, \|\cdot\|_\star)}{n}}$. $\qquad\square$

