# OpenReview forum: "High-Dimensional Calibration from Swap Regret"
_NeurIPS.cc/2025/Conference — NeurIPS 2025 oral_

### Official Review · Reviewer_VeHt · 2025-06-20

**Clarity:** 4
**Significance:** 3
**Originality:** 2
**Rating:** 5
**Confidence:** 5

**Summary:**

The paper considers the problem of online calibration of multidimensional outcomes over an arbitrary convex set $\mathcal{P}$. More formally, at time $t \in [T]$, the forecaster specifies a distribution $x_{t} \in \Delta(\mathcal{P})$, the adversary responds with $y_{t} \in \mathcal{P}$, and the forecaster is penalized by the calibration error measured with respect to a distance function $D$ as $\frac{1}{T}\sum_{p \in \mathcal{P}} \left(\sum_{t = 1} ^ {T} x_{t}(p)\right) D(\nu_{p}, p)$, where $\nu_{p} = \frac{\sum_{t = 1} ^ {T} x_{t}(p) y_{t}}{\sum_{t = 1} ^ {T} x_{t}(p)}$ corresponds to the empirical average of the predictions conditioned on the rounds where the prediction made is $p$. Prior work by Peng'25 proved that $\ell_{1}$-Calibration error (corresponding to $D(p, q) = ||p - q||$) can be achieved after $T = d^{\mathcal{O}(\frac{1}{\varepsilon ^ {2}})}$ rounds when $\mathcal{P} = \Delta_{d}$ is the simplex and $y_{t}$'s correspond to the cannonical basis vectors. Moreover, Peng'25 also showed that $T = d^{\Omega \left(\log \frac{1}{\varepsilon}\right)}$ rounds are necessary to achieve at most $\varepsilon$ $\ell_{1}$-Calibration error. Following up on Peng's work, the current paper does the following:

(1) It proposes an algorithm (TreeCal), which essentially coincides with Peng's algorithm, and establishes a more general result that if OLO (over $\mathcal{P}$) can be achieved in $\mathcal{O}(\sqrt{\rho T})$ external regret, then $\ell_{1}$-Calibration error can be achieved after $\left(\frac{\text{diam}(\mathcal{P})}{\varepsilon}\right) ^ {\mathcal{O}(\frac{\rho}{\varepsilon ^ {2}})}$ rounds. This recovers the result of Peng'25 by setting $\rho = \log d$, which follows from the regret guarantee of Hedge in learning with expert advice.

(2) It improves Peng's lower bound, showing that for $d \ge \text{poly}(\frac{1}{\varepsilon})$, $T = \exp(\text{poly}(\frac{1}{\varepsilon}))$  rounds are necessary to achieve at most $\varepsilon$ $\ell_{1}$-Calibration error. This shows that an exponential dependence on $\frac{1}{\varepsilon}$ is necessary and precludes the existence of bounds of the form $d^{\log \frac{1}{\varepsilon}}$ which were allowed in Peng's work.

**Questions:**

(1) I would appreciate some clarification of the steps spanning lines 993-994 in the appendix.

(2) For this line of work, I find the comparison between the $(\frac{1}{\varepsilon}) ^ {\text{poly}(d)}$ versus $d^{\text{poly}(\frac{1}{\varepsilon}})$ a bit unclear. What is the regime that these bounds are compared against?

(3) As I mentioned in the strengths and weaknesses, the statistical dependence on $\varepsilon$ as a function of $T$ is rather weak (for swap regret minimization, compare this to the $\sqrt{T}$ dependence achievable via the BM reduction versus the $\frac{T}{\log T}$ achieved by the TreeSwap algorithm) and is inevitable since the current papers results build upon that of Daskalakis et al. I understand that the purpose of this paper and Peng's work was really to resolve the exponential dependence on $d$ for $\ell_{1}$-Calibration; however, I am curious if the dependence on $T$ can be improved further, i.e., can the $T^{1 - \frac{1}{d + 1}}$ bound for $\ell_{1}$-Calibration be improved further? I am also curious if the authors have thought about improved lower/upper bounds for $\ell_{2}$-Calibration for $d = 2$.

**Ethical Concerns:**

["NO or VERY MINOR ethics concerns only"]

**Final Justification:**

I did not have any concerns and continue to vote for acceptance.

**Limitations:**

Yes

**Paper Formatting Concerns:**

No Formatting Concerns.

**Quality:**

3

**Strengths And Weaknesses:**

Strengths: (1) The perspective that TreeCal is a special case of the TreeSwap algorithm of Daskalakis et al. is novel to this work. In particular, TreeSwap with Follow-the-Leader as the external algorithm yields the TreeCal algorithm. This perspective allows the authors to obtain a set of more general results for an arbitrary $\mathcal{P}$.

(2) The connection between the achievable bounds on $\ell_{1}$-calibration and the optimal external regret bound of OLO is particularly interesting. More specifically, the TreeCal algorithm does not utilize any OLO algorithm as a subroutine; however, it still achieves a bound that depends on the optimal regret bound of an OLO instance.

(3) A strength of the paper also lies in its simplicity, e.g., the lower bound for $\ell_{1}$-calibration follows by existing results and an observation due to Foster and Vohra, however, it is commendable that the authors connect the above results to improve upon Peng's lower bound. The presentation of the algorithm is exceptionally clear, and the paper can be very well understood from the main body.

Weaknesses: The paper is based on observations that were already known and therefore does not offer significant insights from a technical perspective. However, that being said, tying these observations together to derive meaningful results is a major contribution of this work; therefore, I only consider the technical aspect as a minor weakness. Moreover, some of the extensions of the existing results, e.g., the claim that the algorithm of Daskalakis et al. (with the Follow-the-Leader algorithm as a subroutine) achieves the desired bound is not immediately obvious; the analysis argues that the fictitious Be-the-Leader algorithm achieves a particular guarantee and that the predictions made by Follow-the-Leader and Be-the-Leader are close. This introduces other minor technical nuances in the analysis, e.g., working with a labelled tree.

Overall, I find these results quite interesting and recommend acceptance. On the upper bound side, achieving polynomial dependence on $d$ for multidimensional calibration was open for a long till the recent work by Binghui Peng. The current paper improves our understanding of Peng's algorithm by establishing connections with the recent breakthroughs in Swap Regret minimization. The statistical dependence of the error parameter on the number of rounds is weak; however, this is a manifestation of the improvements in swap regret minimization literature. On the lower bound side, the improvement upon Peng's lower bound is quite strong.

Below is a list of minor typos that I found while reading the main paper and the appendices:

(1) Line 158; the definition of external regret is incorrect.

(2) Line 255; typo in "notation".

(3) Line 258; $\ell$ should range from $1 \le l \le L$.

(4) Line 357; "algorithm $\epsilon$-calibrated"; should be "algorithm producing" $\epsilon$-calibrated predictions.." or something similar;

(5) Line 830; $x - \frac{z_{1}}{2}$ should be $p - \frac{z_{1}}{2}$.

(6) In the equations spanning lines 831 and 832: (a) the second equality should have $-\frac{1}{4} \left \langle \nabla R(\frac{p}{2}), y - p\right \rangle$ (instead of $-\frac{1}{2}$); (b) the second equality should have $p - y$ in the last term; (c) adjust the constant $3\rho$ accordingly.

(7) Line 856; "even in the even" sounds incorrect.

(8) Line 860; should be $\mathcal{P} \times [0, H]^\star$ instead.

(9) Line 965; fix the "??"

---

> ### Author Rebuttal · Authors · 2025-07-31
>
> Thank you for your detailed review. To answer your questions:
>
> (1) We remark that there is a typo in the first line of the proof, it should say $\phi : B_1^d \to \Delta^{2d+1}$. The first equality below line 993 is the definition of calibration error. The following inequality uses the fact that $x_t’(p’) = \sum_{p : \psi(p) = p’} x_t(p)$ (by the definition of $x_t’$ as the pushforward of $x_t$ by $\psi$) together with Jensen’s inequality to split apart the norm into the individual pieces corresponding to the values of $p$ for which $\psi(p) = p’$. The final inequality uses the fact that $\psi$ is a $\ell_1$-contraction on $\Delta^{2d+1}$.
>
> (2) We are interested in the regime when d is large and epsilon is not too large (e.g., think of eps as a constant or $\epsilon \sim 1/\text{polylog}(d)$). For such parameters, $(1/\epsilon)^d$ is extremely large, and in particular larger than $d^{1/\epsilon}$.
>
> (3) Those are great questions for future work!

---

> > ### Comment · Reviewer_VeHt · 2025-08-06
> > **Thanks for the response**
> >
> > I thank the authors for their responses. I maintain my positive evaluation of the work; however, I encourage the authors to fix the typos in the subsequent revision.

---

### Official Review · Reviewer_zPv9 · 2025-06-30

**Clarity:** 3
**Significance:** 3
**Originality:** 2
**Rating:** 4
**Confidence:** 4

**Summary:**

This paper studies online calibration when the events and forecasts live in a high-dimensional space. The authors show that if $\rho$ is the rate of OLO under a convex set with certain norm, then there is a simple forecasting algorithm that achieves $(diam(P)/\epsilon)^{O(\rho/\epsilon)}$ calibration error. This result recovers [Peng 25] when norm is L-1 and set is d-dimensional simplex. They also establish a lower bound that improves the lower bound of Peng 25.

The technical approach consists of two main components:
1. Following [LSS25, KFO+25], the authors connect calibration (with bregman divergence as distance measure) to full swap regret with losses chosen as bregman divergence from the events $x_t$. on the one hand, this connection allows them to inherit the guarantees from the TreeSwap algorithm [DDFG24]. On the other hand, it provides an upper bound on the calibration error when using squared norm as the distance measure.
2. The authors show that the TreeCal algorithm [Peng25] can be recovered as a special case of TreeSwap with FTL as the base algorithm. They provide a novel and simplified analysis of TreeCal's calibration bounds by examining its difference from TreeSwap with BTL, resulting in a clean analysis that works for arbitrary norms and prediction sets.

**Questions:**

Please see strengths and weaknesses.

**Ethical Concerns:**

["NO or VERY MINOR ethics concerns only"]

**Final Justification:**

I appreciate the contributions of this paper and am happy to recommend acceptance. The reason for not giving a higher score is that the paper mainly brings together results from different papers rather than introducing new technical tools. That said, I appreciate the simplified and unified analysis it provides, and I believe this paper makes a nice contribution.

**Limitations:**

yes

**Quality:**

3

**Strengths And Weaknesses:**

Strengths:

The paper provides a simplified and unified analysis framework for high-dimensional calibration under arbitrary prediction sets and norms, offering cleaner proofs compared to Peng25. The proposed algorithm achieves calibration without requiring knowledge of the OLO rate or diameter of the prediction set, making it particularly appealing. The new analysis also removes the need to mix the prediction with uniform distribution (as Peng25 did).

Weaknesses:

While elegant, the connection between calibration and OLO rates largely combines existing results rather than providing new technical tools: The reduction from calibration (using Bregman divergence) to swap regret builds directly on prior work [LSS25,KFO25], the swap regret part invokes results from [DDFG24], the analysis through BTL and FTL are hinted by [Peng25] (although this paper provides a more general proof that works for any norm and also removes the need of uniform distribution as stablizer), and the connection to OLO rates also directly build on [GSJ24]. That said, the paper brings these separate techniques together into a more unified analysis.

---

> ### Author Rebuttal · Authors · 2025-07-31
>
> Thank you for your positive feedback.

---

### Official Review · Reviewer_EHNL · 2025-07-03

**Clarity:** 3
**Significance:** 3
**Originality:** 3
**Rating:** 5
**Confidence:** 3

**Summary:**

The paper studies an online calibration problem. Unlike previous works which mostly focus on 1-dim online calibration, this work focuses on a high-dimensional setting where the goal here is to produce a sequence of multi-dimensional forecasts that minimizes the calibration error.

The main result of is an online algorithm that can guarantee vanishing calibration error relative to an arbitrary norm.

**Questions:**

N/A

**Ethical Concerns:**

["NO or VERY MINOR ethics concerns only"]

**Quality:**

3

**Strengths And Weaknesses:**

Online high-dimensional calibration is known to be a challenging problem (as the generalization of typical online calibration methods require an exponential number of rounds w.r.t. the dimension). Recently, the work [Pen25] has shown that it is possible to achieve a calibration error rate $\epsilon T$ in $d^{O(\ln 1/\epsilon)}$ rounds when focusing on $\ell_1$-norm calibration error. This work advances this result with being able to generalize the result for arbitrary $\|\cdot\|$ error. They achieve this with a more simplified and a unified analysis.

I do not have major concern of this work. This work is a theoretical work and I believe that this work would be a good addition to NeurIPS.

Maybe this question is beyond the scope of this work, but it seems that the exponential  $\ln(1/\epsilon)$-dependency in the lower bound not satisfying, would it be possible to break this barrier by considering a smoothed adversary?

Minor points:

1. Line 73, I guess you are referring to Theorem 1.1?
2. Line 188, where is Figure 3?

---

> ### Author Rebuttal · Authors · 2025-07-31
>
> Thank you for your positive feedback. The question regarding a smoothed adversary is a good question for future work.
>
> Regarding line 188: Figure 3 is in the supplemental material (we will correct that point in the final version).

---

### Official Review · Reviewer_KJ55 · 2025-07-05

**Clarity:** 3
**Significance:** 3
**Originality:** 2
**Rating:** 5
**Confidence:** 3

**Summary:**

The paper consdiers the problem of issuing calibrated forecasts in d-dimensional Euclidean space.  It develops the TreeCal algorithm, and analyses its average calibration error as function of the number of rounds. The proof goes through swap regret. A lower bound is also provided. Lower and upper bounds are exponentials of polynomials in 1/epsilon, though with different powers.

**Questions:**

I am to some degree surprised by the simplicity of the eventual algorithm, and the fact that it is independent of several problem parameters including the Bregman generator and its radius. These are all only required in the analysis. Do the authors understand why it is the case, and would it be plausible that there exists an even simpler direct analysis?

**Ethical Concerns:**

["NO or VERY MINOR ethics concerns only"]

**Final Justification:**

I thank the authors for clarifying that indeed the domain does come into the algorithm, and not just the analysis.

**Limitations:**

yes

**Paper Formatting Concerns:**

-

**Quality:**

3

**Strengths And Weaknesses:**

Quality
I went over the proofs of the main result Section 3, and these are all good.

Clarity
The paper is written with the reader in mind. At several points does it immediately address questions the reader may have with a remark or footnote, after raising these questions by presenting results. I'm thinking of repeated actions, ranges of Bregman divergences vs their generators, loss vs parameter distance, etc. This makes the paper a pleasant read.

Significance
The improvement in lower and upper bounds for d-dimensional calibration is significant. The method is actually easy to implement, and the contribution of the paper is its design and analysis.

Originality
The originality lies in the analysis. The algorithm itself was already proposed before. The new perspective taken here allows a different bounding technique.

---

> ### Author Rebuttal · Authors · 2025-07-31
>
> Thank you for your positive review. We are not aware of a simpler proof of our main result. Note that while the Bregman regularizer R does not explicitly show up in the final bound, in the definition of Rate the regularizer R is chosen to solve an optimization problem involving the given norm. So the final bound takes the geometry of $\mathcal{P}$ (as well as the norm) into play.

---

### Official Review · Reviewer_Fhoa · 2025-07-13

**Clarity:** 2
**Significance:** 4
**Originality:** 4
**Rating:** 6
**Confidence:** 5

**Summary:**

The paper studies the online calibration problem in high dimension setting. In the task of online calibration, there is a sequence of $T$ days and the algorithm needs to make a prediction $p_t$ at the beginning of each day $t$, then the nature (or the adversary) picks an outcome $i_t\in [d]$. The hope is that the algorithm is calibrated, roughly meaning that on average, the time that it predicts $p$, the empirical average is also $p$. The calibration error is mostly measured in $\ell_1$ distance in the literature, but other distance metric is well motivated as well.


The main contribution are two folds:

(1) On the algorithmic side, the paper generalizes the previous work of (Peng 2025) and gives a generic reduction from swap regret / external regret to calibration. In particular, the paper shows that whenever there exists an online linear optimization algorithm that has $O(\sqrt{\rho T})$ worst case regret, then for the corresponding calibration problem, it gives an $\epsilon$-calibrated algorithm after $T = \exp(O(\rho/\epsilon^2))$. For the most studied $\ell_1$ calibration, this matches the previous algorithm of (Peng 2025), for $\ell_2$ calibration, it gives a new results that otabin $\eps$-$\ell_2$ calibration in $(1/\epsilon)^{O(1/\epsilon^2)}$ days.

(2) On the lower bound side, the algorithm gives improved lower bound, showing that any online calibration algorithm that guarantees $\eps$ $\ell_1$ calibration error requires $T = \min \{\exp(\poly(d), \poly(\epsilon))\}$ days, this improves the previous lower bound of $d^{\Omega(\log(1/\epsilon))}$ of (Peng 2025). The lower bound is obtained via a smart reduction to the previous generic lower bound for swap regret minimization.

**Questions:**

.

**Ethical Concerns:**

["NO or VERY MINOR ethics concerns only"]

**Limitations:**

.

**Paper Formatting Concerns:**

.

**Quality:**

4

**Strengths And Weaknesses:**

The reduction is very nice and solves long standing open questions. The lower bound is also very nice and smart. I strongly recommend acceptance to NeurIPS.


There are minor suggestions:

(1) It would be good to mention that the approach of (Peng 2025) is also inspired by swap regret minimization.

(2) There is a discussion between distributional calibration and calibration in the appendix, the distinction might seems minor to the authors, but for me (and maybe other readers), then distinction is non-negligible (e.g. in Foster 97, if two forecasters are distributional calibrated, then playing "best response" might not lead to correlated equilibrium?). Why not put the discussion earlier. Also, it would be good the mention that the lower bound only holds when the outcome $i_t$ is some distribution but not a single outcome (correct me if I am wrong.)

---

> ### Author Rebuttal · Authors · 2025-07-31
>
> Thank you for your positive feedback and for the helpful suggestions! We will take them into account when preparing the final version of the paper.

---

### Decision · Program_Chairs · 2025-09-17

**Decision:**

Accept (oral)

**Comment:**

The paper considers the problem of high-dimensional online calibration. The recent work of Peng'25 showed a very surprising result that d-dimensional L1 calibration to error eps is achievable in d^(1/eps^2) steps. This is surprising, since in d dimensions the discretization of the simplex is of size exponential in d, and previous approaches required exp(d) steps. This paper recovers, and also significantly improves on this result from Peng'25 in many ways. The paper shows a bound which extends to other metrics beyond L1, such as L2 distance. Interestingly, the same algorithm can simultaneously achieve low calibration error simultaneously for all multiple norms. The paper gives a generic reduction from swap regret / external regret to calibration, and shows that the TreeCal algorithm introduced in Peng'25 is a special case of the earlier TreeSwap algorithm. This perspective on calibration is novel, significant and impactful. It greatly clarifies and helps understand the landscape of algorithms, and relations between different problems in this space. The paper also shows a stronger lower bound, showing that exp(poly(1/eps)) rounds are necessary, an exponential improvement over the lower bound in Peng'25.

The reviewers all found the paper to be very well written. It gives the reader a good lay of the land with respect to previous results, and also provides sufficient intuition for the results.

I believe that the results here are highly significant and of significant interest to a broad audience, and hence recommend that the paper be selected for an oral presentation. An oral presentation would give the broad Neurips audience exposure to recent advances in calibration, including the strong connections to swap regret pointed out in this work. I also encourage the authors to also use the presentation as an opportunity to introduce the work of Peng'25 to the audience.